# *VIP*: Visual-guided Prompt Evolution for Efficient Dense Vision-Language Inference

**Hao Zhu** [* 1 2]  **Shuo Jin** [* 3 4]  **Wenbin Liao** [1 2]  **Jiayu Xiao** [1 2]  **Yan Zhu** [2]  **Siyue Yu** [3]  **Feng Dai** [1]

## Abstract

Pursuing training-free open-vocabulary semantic segmentation in an efficient and generalizable manner remains challenging due to the deep-seated spatial bias in CLIP. To overcome the limitations of existing solutions, this work moves beyond the CLIP-based paradigm and harnesses the recent spatially-aware dino.txt framework to facilitate more efficient and high-quality dense prediction. While dino.txt exhibits robust spatial awareness, we find that the semantic ambiguity of text queries gives rise to severe mismatch within its dense cross-modal interactions. To address this, we introduce **VI**sual-guided **P**rompt evolution (*VIP*) to rectify the semantic expressiveness of text queries in dino.txt, unleashing its potential for fine-grained object perception. Towards this end, *VIP* integrates alias expansion with a visual-guided distillation mechanism to mine valuable semantic cues, which are robustly aggregated in a saliency-aware manner to yield a high-fidelity prediction. Extensive evaluations demonstrate that *VIP*: ❶ surpasses the top-leading methods by $1.4\% \sim 8.4\%$ average mIoU, ❷ generalizes well to diverse challenging domains, and ❸ requires marginal inference time and memory overhead. Our code is publicly available at GitHub ⭕.

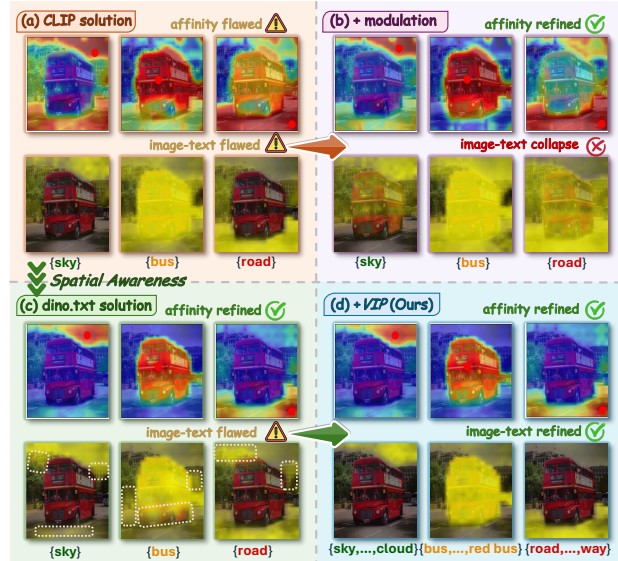

*Figure 1.* **Comparison of image affinity and dense image-text activation.** (a) Prevalent CLIP-based *one-layer* attention modulation methods. (b) Extended *two-layers* modulation on CLIP-based methods to further rectify spatial bias. (c) Spatially-aware *dino.txt* solution. (d) Ours *VIP*, building upon *dino.txt*.

## 1. Introduction

The progress in vision-language models (VLMs) (Radford et al., 2021; Jia et al., 2021; Li et al., 2023; Jiang et al., 2026) provides an effective solution for open-vocabulary semantic segmentation (OVSS), which aims to segment images

into arbitrary categories specified by text prompts. Conventional approaches address OVSS by fine-tuning VLM (Liang et al., 2023; Wu et al., 2024; Xu et al., 2023; Xing et al., 2023) or training auxiliary modules (Ding et al., 2023; Cha et al., 2023; Lin et al., 2023; He et al., 2025) to equip the model with spatial awareness. Despite significant strides, this paradigm relies heavily on manual annotations and time-consuming training to adapt VLM.

Motivated by this, recent studies have been drawn to exploring the potential of CLIP (Radford et al., 2021) to perform OVSS in a fully *training-free* manner. Given that CLIP aligns global image representations with text embeddings during pre-training, the dense image features inherently lack fine-grained spatial awareness. To mitigate this, the common practice is to modulate the self-attention mechanism in the last layer of the *image encoder* to render the image features more discriminative (Wang et al., 2024; Lan et al., 2024b; Kang et al., 2025; Chi et al., 2025). However, the deep-seated spatial bias induced by pre-training

---

[*]Equal contribution  [1]Institute of Computing Technology, Chinese Academy of Sciences, Beijing, China [2]School of Computer Science and Technology, University of Chinese Academy of Sciences, Beijing, China [3]XJTLU [4]University of Liverpool. Correspondence to: Feng Dai <fdai@ict.ac.cn>.

*Proceedings of the 43$^{rd}$ International Conference on Machine Learning*, Seoul, South Korea. PMLR 306, 2026. Copyright 2026 by the author(s).

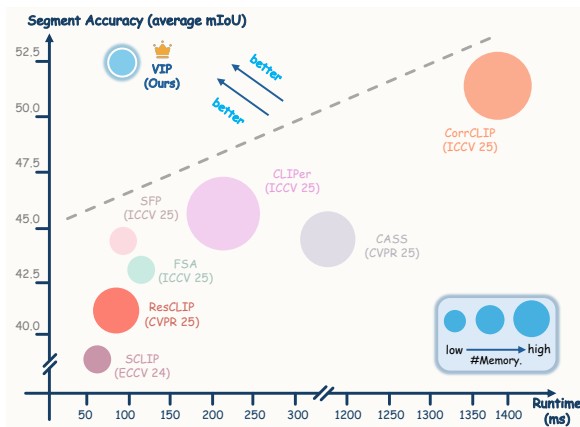

*Figure 2.* **Comparisons of segmentation accuracy and inference latency.** Our *VIP* establishes a new state-of-the-art for this field.

impedes CLIP-based methods from effectively restoring the spatial awareness, as extended modulation often disrupts cross-modal alignment, thereby confining prevalent approaches to suboptimal performance, as shown in Figure 1(a)-(b). To remedy this, a few approaches (Shi et al., 2025; Zhang et al., 2025) resort to using the Segment Anything Model (SAM) (Kirillov et al., 2023; Ravi et al., 2025) as a post-processor to alleviate the performance bottleneck of CLIP-based methods. Though effective, they introduce significant inference latency and memory overhead, hampering their applicability in many real-world applications. Moreover, the improvements achieved by such methods are essentially rooted in SAM's large-scale training with labor-intensive *mask annotations*, and may not generalize well to out-of-distribution domains, *e.g.*, remote sensing imagery.

On the other hand, the success of DINO (Caron et al., 2021) has inspired recent works (Naeem et al., 2024; Maninis et al., 2025; Tschannen et al., 2025; Jose et al., 2025) to incorporate self-supervised objectives in the pre-training of VLMs to instill spatial awareness without mask supervision. Among them, dino.txt (Jose et al., 2025) stands out for its superior performance on both global and dense visual understanding tasks. By virtue of grounding in the frozen DINOv3 backbone (Siméoni et al., 2025), the image representations of dino.txt are naturally endowed with robust local discriminability. Crucially, with a simple self-correction, dino.txt can thoroughly eliminate the spatial bias induced by global text alignment pre-training, without shifting image features away from the learned cross-modal feature distribution, as shown in Figure 1(c). *Driven by this, our work moves away from prior CLIP-based paradigms plagued by spatial bias, and instead, pivots to spatially-aware dino.txt in pursuit of efficient and high-fidelity dense inference.*

Although dino.txt exhibits powerful visual discriminability, however, as a task-agnostic pre-trained VLM, it often mani-

fests dense cross-modal *mismatch* when directly applied to downstream OVSS. The prevailing paradigm typically leverages text queries, derived from benchmark-defined *canonical class names*, as semantic probes to activate target regions within dense image features. However, there is a lexical gap between the canonical class names and the image-caption descriptions found in the pre-training data of dino.txt (Jose et al., 2025). Furthermore, a single canonical name has limited semantic capacity and may fail to encapsulate the visual diversity of an object category. As a result, this semantic ambiguity of text queries hinders the spatially-aware image features from accurately responding to semantic anchors, marked as orange rectangles at the bottom of Figure 1(c).

To mitigate this limitation, we propose a simple yet effective solution, termed **VI**sual-guided **P**rompt evolution (*VIP*), which refines the semantic expressiveness of text queries in dino.txt and facilitates high-quality object activations. Specifically, there are three key aspects of *VIP*: ***First***, we employ a large language model to generate a candidate pool of image-caption style aliases for each class name. While these aliases enhance semantic coverage, they may blur the separability of class boundaries, leading to erroneous activations. ***Second***, we thus devise a visual-guided alias distillation scheme that exploits hierarchical visual priors to automatically mine valid aliases from the candidate pool. ***Third***, we introduce a saliency-aware soft aggregation technique to robustly integrate activation maps derived from multiple filtered names within the same category into a unified prediction. In addition, we explore the transferability of our core idea to text templates, which can further enhance the perceptual capability of text queries. Our method significantly narrows the gap between dense visual representations and text queries, thereby yielding more precise dense cross-modal activations, as depicted in Figure 1(d).

We conduct comprehensive evaluations on multiple benchmarks, covering diverse scenarios in natural images, urban street scenes, and remote sensing imagery. Empirical results demonstrate that *VIP* is ❶ **top performing**, surpassing existing state-of-the-arts (SOTA) by $1.4\% \sim 8.4\%$ average mIoU on standard benchmarks; ❷ **versatile generalization**, consistently delivering optimal results across diverse challenging domains; ❸ **high efficiency**, operating with only $7\%$ inference time and $50\%$ memory overhead compared to recent SAM-based SOTA, as illustrated in Figure 2.

In sum, we make the following key contributions:

① We identify that existing training-free OVSS solutions grounded in CLIP suffer from intractable spatial bias. In light of this, we shift our focus toward the spatially-aware dino.txt paradigm to circumvent the performance bottleneck in prior works.

② To alleviate the dense cross-modal mismatch in

dino.txt, we propose **VI**sual-guided **P**rompt evolution (*VIP*), which rectifies the semantic expressiveness of text queries and effectively unlocks model's fine-grained object perception capability.

③ Extensive evaluations demonstrate that our *VIP* achieves SOTA performance in diverse domains with excellent computational efficiency.

## 2. Related Work

**Vision-language Models.** The advent of CLIP (Radford et al., 2021) has catalyzed extensive exploration into vision–language models (VLMs) in the vision community. Building upon this foundation, follow-up works such as ALIGN (Jia et al., 2021), CoCa (Yu et al., 2022), OpenCLIP (Cherti et al., 2023), SigLIP (Zhai et al., 2023) and InternVL (Chen et al., 2024) introduce significant advances in data curation and learning objectives. However, most existing VLMs focus on learning global image-text alignment that overlook the performance on downstream dense prediction tasks. To address this, several methods (Naeem et al., 2024; Maninis et al., 2025; Tschannen et al., 2025) attempt to integrate DINO-style (Caron et al., 2021; Oquab et al., 2024) self-supervised objectives into the pre-training stage of VLMs, aiming to enhance the model's spatial perception without any manual annotations. In this setting, dino.txt (Jose et al., 2025) applies contrastive learning to align the text representations with the frozen dense image features of DINOv3 (Siméoni et al., 2025), demonstrating exceptional performance on both image-level and pixel-level visual understanding tasks. In this work, we build on top of dino.txt framework to liberate the training-free OVSS model from the intrinsic spatial bias embedded in CLIP, thereby facilitating efficient and high-quality dense prediction.

**Training-free Open-vocabulary Semantic Segmentation.** The *training-free* OVSS paradigm performs text-specific semantic segmentation by directly aligning text embeddings with image patch embeddings derived from the frozen VLM. Prevailing solutions (Zhou et al., 2022; Wang et al., 2024; Li et al., 2025b; Kang et al., 2025; Chi et al., 2025) are devoted to modulating the self-attention mechanism in the last layer of image encoder to enhance CLIP's spatial awareness, while several methods (Jin et al., 2025; Shao et al., 2024; Bai et al., 2025) mitigate the spatial defects by rectifying the outlier tokens. Following this line of work, a few recent efforts (Kang & Cho, 2024; Lan et al., 2024b; Sun et al., 2025; Kim et al., 2025; Stojnić et al., 2025) incorporate extra off-the-shelf vision foundation models, *i.e.*, DINO (Caron et al., 2021; Oquab et al., 2024) and Stable Diffusion (Rombach et al., 2022), to further improve the robustness of attention matrices. Some methods (Shi et al., 2025; Zhang et al., 2025) additionally prompt SAM (Kirillov et al., 2023; Ravi

et al., 2025) to generate high-quality masks that serve to refine CLIP's dense prediction. Nevertheless, the above works predominantly target the image modality, leaving the potential of the textual modality largely untapped. Closer to our work, (Sun et al., 2024; Benigmim et al., 2025) explicitly investigate the role of the textual modality. Specifically, (Sun et al., 2024) expands each class name into 10 synonyms and fuses the corresponding text representations, and (Benigmim et al., 2025) identifies *class-experts* from 80 handcrafted CLIP text templates. However, the inherent bottlenecks of CLIP's spatial discriminability curtail the efficacy of these textual improvements, resulting in only marginal performance gains. In contrast, our core idea, *i.e.*, exploiting robust visual priors to refine the semantic expressiveness of text queries, is conceptually novel and rarely explored before. It is executed under a fully training-free manner without any mask annotations, while entailing negligible inference latency during deployment.

**Training-free VLMs Text Prompts Refinement via Large Language Models.** Leveraging large language models (LLMs) to enrich textual inputs for VLMs has been proven to improve the performance on the zero-shot classification (Menon & Vondrick, 2023; Dong et al., 2025). The core strategy is to query GPT-3 (Brown et al., 2020) to generate class-specific contextual descriptors, thereby deriving more informative text representations (Pratt et al., 2023; Roth et al., 2023; Mirza et al., 2024). In addition, ZPE (Allingham et al., 2023) introduces a prompt ensemble method that assigns ensemble weights to prompt templates by accessing the pre-training data of VLM. While these approaches have advanced zero-shot image classification, these gains fail to extend to dense prediction. Different from prior works, we introduce an optimization-free textual refinement strategy to improve the pixel-level perception capability of VLM.

## 3. Method

In this section, we commence by briefly reviewing the mechanism of dino.txt (Jose et al., 2025) for dense prediction (§3.1). The following three parts represent our proposed method (*cf.* Figure 3): §3.2 LLM-driven class semantic expansion, §3.3 visual-guided alias distillation, and §3.4 saliency-aware soft aggregation. Finally, we present the transferability of our method to the text templates in §3.5.

### 3.1. Preliminary

Akin to CLIP (Radford et al., 2021), dino.txt (Jose et al., 2025) employs the dual-tower framework to independently encode image and text inputs, and achieves cross-modal alignment through the contrastive objective. The crucial distinction lies in the treatment of the visual modality: while CLIP trains its image encoder from scratch, dino.txt builds upon the spatially-aware image representations derived from

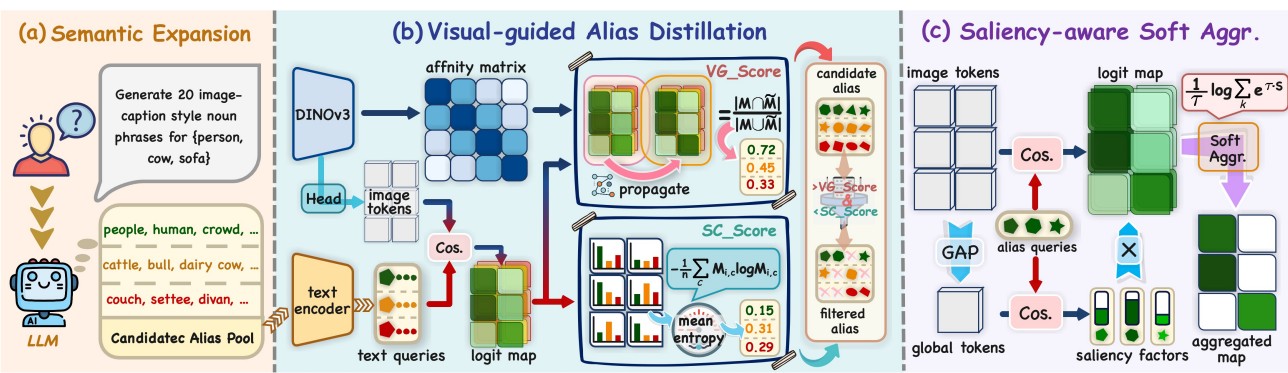

*Figure 3.* **Overview of our proposed *VIP*.** It comprises three key modules: §3.2 **semantic expansion**, §3.3 **alias distillation**, and §3.4 **activation aggregation**, to refine the semantic expressiveness of text queries. Here, the distinct colored shapes ●●● represent text queries from different categories, and different shapes within the same color ●●★ denote text queries of the same category.

DINOv3 (Siméoni et al., 2025). Specifically, the image encoder of dino.txt consists of a frozen DINOv3 ViT (Dosovitskiy et al., 2021) backbone and two-layer transformer blocks (we refer to "adapter"). We denote the image patch tokens output by the backbone and the adapter as $V \in \mathbb{R}^{hw \times d}$ and $I \in \mathbb{R}^{hw \times d}$, respectively, where $hw$ represent the number of patch tokens and $d$ is the feature dimension. Here we omit the [CLS] token for brevity. During pre-training, dino.txt applies global average pooling to $I$ and aligns it with the text embedding via contrastive learning. Therefore, in the inference phase, we can obtain the segmentation logits map $\mathcal{M}$ by directly calculating the cosine similarity between patch tokens $I$ and text queries $T \in \mathbb{R}^{C \times d}$:

$$\mathcal{M} = \text{cosine}(I, T), \quad \mathcal{M} \in \mathbb{R}^{hw \times C}, \quad (1)$$

where $C$ is the number of categories for a given dataset. The final segmentation output is derived by applying the $\text{argmax}$ operation to the logit map $\mathcal{M}$ that assigns the most probable class to each pixel. By default, we consider that $\mathcal{M}$ is normalized through $\text{softmax}$ function.

**Self-correction.** While dino.txt's global alignment also dilutes spatial awareness originating from DINOv3, such cues remain latent rather than vanishing entirely. Through a simple self-correction strategy, we can effectively disentangle the spatial bias caused by image-level pre-training. We leverage the self-attention modulation technique adopted in previous CLIP-based approaches (Lan et al., 2024b) to achieve self-correction. In particular, we replace the attention weights in vanilla self-attention mechanism with the self-similarity derived from DINOv3 output patch tokens $V$, aiming to restore the inherent spatial awareness while maintaining the semantic coherence:

$$Attn = \text{softmax}(\frac{VV^\top}{\sqrt{d}}), \ Attn \in \mathbb{R}^{hw \times hw}. \quad (2)$$

This modification is applied to the two-layer transformer blocks of the adapter to thoroughly reawaken the capacity for fine-grained object perception of image tokens $I$.

**Limitations.** Crucially, unlike CLIP-based approaches, restoring spatial awareness in dino.txt image tokens does not disrupt the well-established unified feature space (see §4.3 for more details). This is mainly due to the fact that dino.txt freezes the DINOv3 backbone and optimizes the adapter solely, thereby preventing the image tokens from being severely compromised. However, although dino.txt can fundamentally eliminate spatial bias via self-correction, its segmentation results remain sub-optimal, even falling behind several CLIP-based solutions. It reveals that the bottleneck of dino.txt lies not only in the discriminative power of the visual modality, but rather in the ***cross-modal mismatch***, where rich visual features fail to find their precise semantic anchors in the text embedding space. Driven by this insight, our work pivots to the text modality, especially targeting the lexical enrichment of semantic anchors to bridge the cross-modal semantic gap in dino.txt framework.

### 3.2. Semantic Expansion by Large Language Model

Prevailing training-free OVSS methods generally employ the canonical class names defined by datasets as semantic anchors to retrieve visual objects. However, a solitary class name is semantically sparse and may fail to capture the intrinsic visual diversity associated with an object category. Furthermore, there exists a lexical discrepancy between the canonical class names defined by benchmarks and the natural language descriptions encountered during dino.txt's pre-training (Jose et al., 2025). For example, the term "*people*" is more prevalent in image captions rather than the canonical category label "*person*". Ultimately, these limitations prevent the rich and spatially-aware dense visual representations from aligning with semantic text anchors.

To mitigate these limitations, we automatically construct a set of candidate aliases by prompting a large language model (LLM), to extend the semantic coverage of each category label. Specifically, we query the LLM with the following prompt format:

**Query:** Generate 20 image-caption style noun phrases for {category name}, covering synonyms, plurals, hyponyms, and concrete visual variants.

Here, {category name} is substituted for a given canonical class name $c$. We provide more implementation details in the Appendix A.1. Figure 3(a) presents several examples of class aliases generated by the LLM. These aliases typically encompass a wide range of linguistic forms, including synonyms and hyponyms, thereby enriching the semantic representation of the abstract category labels.

**Challenge.** Nevertheless, it is crucial to acknowledge that naively integrating the LLM-generated aliases is not a silver bullet for bridging the cross-modal gap. Instead, it presents a double-edged sword: unchecked semantic expansion may obscure the discriminative boundaries among categories, thereby compromising segmentation accuracy.

### 3.3. Visual-guided Alias Distillation

To address the above challenge, we introduce a visual-guided alias distillation strategy to automatically filter candidate aliases *without* relying on any mask annotations. Our key insight lies in exploiting high-fidelity visual cues as an implicit supervision signal, which scores generated aliases based on their visual-text alignment. To this end, we first extract multi-layer attention matrices from the DINOv3 backbone to incorporate hierarchical visual priors as guidance. We formulate the aggregated visual affinity as:

$$\mathcal{A} = \frac{1}{L} \sum_l \tilde{Attn}_l, \ \mathcal{A} \in \mathbb{R}^{hw \times hw}, \tag{3}$$

where $\tilde{Attn}_l$ denotes the $l$-th layer attention matrix derived from the DINOv3 backbone, and $L$ is the number of layers. The affinity matrix $\mathcal{A}$ encapsulates the hierarchical visual guidance encoded by DINOv3, capturing both low-level boundary perception and high-level semantic cues.

**Visual Grounding Score.** Inspired by this, we treat the pairwise affinity $\mathcal{A}$ as the annotation-free visual supervision and leverage the image-text activation map $\mathcal{M}$ (*cf.* Eq. 1) as the *evaluation proxy* to assess the quality of the candidate aliases. In particular, we score each alias by evaluating the consistency between its activation map and visual affinity. Given the dimensionality mismatch between activation maps and visual affinity, we employ an intermediary to evaluate their consistency. To this end, we first adopt random walk algorithm (Vernaza & Chandraker, 2017) to robustly propagate the pairwise affinity on the activation map.

$$\mathcal{W} = \mathcal{Q}^{-1}\mathcal{A}^\alpha, \quad \text{where } \mathcal{Q}_{ii} = \sum_j \mathcal{A}_{ij}^\alpha, \\ \tilde{\mathcal{M}} = \mathcal{W}^\beta \cdot \mathcal{M}, \tag{4}$$

here $\alpha \geq 1$ serves to dampen irrelevant affinities in $\mathcal{A}$, $\beta$ is the number of iterations. For each candidate alias, we

substitute the canonical class name with the alias and derive corresponding logit maps $\mathcal{M}$ and $\tilde{\mathcal{M}}$ for it. By this way, $\tilde{\mathcal{M}}$ implicitly encodes the structural priors inherent in the visual model. Therefore, we compute the *IoU* between $\mathcal{M}$ and $\tilde{\mathcal{M}}$ as *visual grounding score* metric to measure the congruence between alias semantic responses and latent visual priors:

$$\mathtt{VG\_Score} = \mathbb{E}_{\mathtt{img} \sim \mathcal{D}} \Big[ \frac{|\mathcal{M}_c \odot \tilde{\mathcal{M}}_c|}{|\mathcal{M}_c| + |\tilde{\mathcal{M}}_c| - |\mathcal{M}_c \odot \tilde{\mathcal{M}}_c|} \Big], \tag{5}$$

where $\mathcal{D}$ represent the test set, $c$ is the index of alias's class, $|\cdot|$ denote the global summation over all map locations and $\odot$ represents the Hadamard product. The final score for an alias is computed as the average across the test dataset. Note that we only account for scores within the high-activation patches of the alias ($\mathcal{M}_{i,c} \geq 0.4$, where $i$ denotes the patch index). If a given alias fails to manifest a high-activation patch within a image sample, that sample is excluded from this alias's computation. The $\mathtt{VG\_Score}$ enables us to measure the congruence between alias and visual priors, serving as a strong basis for assessing the alias's geometric fidelity.

**Semantic Certainty Score.** However, the $\mathtt{VG\_Score}$ fails to capture the inter-class discriminability, where an alias may exhibit superior visual fidelity yet suffer from semantic ambiguity at the decision boundaries. For this reason, we introduce *semantic certainty score* to evaluate the inter-class semantic discriminability of aliases, serving as the orthogonal counterpart to $\mathtt{VG\_Score}$. We employ *entropy* to implement $\mathtt{SC\_Score}$, as it is a widely recognized metric for quantifying uncertainty, as follows:

$$\mathtt{SC\_Score} = \mathbb{E}_{\mathtt{img} \sim \mathcal{D}} \Big[ -\frac{1}{n} \sum_i^n \sum_c (\mathcal{M}_{i,c} \odot \log \mathcal{M}_{i,c}) \Big], \tag{6}$$

where $n$ represents the number of patches in high-activation regions, and we only consider the entropy values of the patches within this specific region. The $\mathtt{SC\_Score}$ forms a synergistic alliance with the $\mathtt{VG\_Score}$, ensuring that the selected category aliases are both visually aligned and semantically discriminative.

**Automated Alias Filtering Mechanism.** We use the score from canonical class name as an anchor to filter candidate aliases. Specifically, we retain only candidate aliases that achieve a *higher* $\mathtt{VG\_Score}$ but a *lower* $\mathtt{SC\_Score}$ than the corresponding canonical class name. This balanced mechanism facilitates the exploration of semantic diversity while mitigating potential noise given the absence of manual supervision. It also avoids the need for manually tuning hard thresholds and hyperparameters.

### 3.4. Saliency-aware Soft Aggregation

After building the robust alias library, we now focus on *aggregating* alias activation maps within each category into

*Table 1.* **Comparison with the state-of-the-art training-free OVSS methods** on natural images and urban scenes. Best results are in **bold**, and second best are underlined. The inference time and memory footprint of the model are evaluated on the Pascal VOC dataset.

| Methods | Extra Backbone | With Background | | | Without Background | | | | | Avg.↑ (mIoU) | Time.↓ (ms) | Mem.↓ (MB) |
|---|---|---|---|---|---|---|---|---|---|---|---|---|
| | | VOC21 | PC60 | Object | VOC20 | PC59 | Stuff | City | ADE | | | |
| *Dense image-text inference with multi vision models* | | | | | | | | | | | | |
| LaVG [ECCV24] | *DINO* | 61.8 | 31.5 | 33.3 | 81.9 | 34.6 | 22.8 | 25.0 | 14.8 | 38.2 | 140 | 1750 |
| ProxyCLIP [ECCV24] | *DINO* | 61.3 | 35.3 | 37.5 | 80.3 | 39.1 | 26.5 | 38.1 | 20.2 | 42.3 | 105 | 1600 |
| LPOSS [CVPR25] | *DINO* | 62.4 | 35.4 | 34.3 | 79.3 | 38.6 | 26.5 | 37.9 | 22.3 | 42.1 | - | - |
| CASS [CVPR25] | *DINO* | 65.8 | 36.7 | 37.8 | 87.8 | 40.2 | 26.7 | 39.4 | 20.4 | 44.4 | 440 | 2890 |
| FSA [ICCV25] | *DINO* | 63.7 | 36.1 | 38.0 | 82.3 | 39.9 | 27.0 | 38.8 | 20.5 | 43.3 | 110 | 1650 |
| CLIPer [ICCV25] | *Stable Diffusion* | 66.5 | 38.3 | 40.0 | 86.0 | 42.4 | 28.6 | 38.7 | 22.0 | 45.3 | 180 | 3390 |
| Trident [ICCV25] | *DINO+SAM* | 67.1 | 38.6 | 41.1 | 84.5 | 42.2 | 28.3 | 42.9 | 21.9 | 45.8 | 120 | 3710 |
| CorrCLIP [ICCV25] | *DINO+SAM2* | 74.8 | 44.2 | 43.7 | 88.8 | 48.8 | 31.6 | 49.4 | 26.9 | 51.0 | 1440 | 3200 |
| *Dense image-text inference with single vision model* | | | | | | | | | | | | |
| CLIP [ICML21] | ✗ | 16.2 | 7.7 | 5.5 | 41.8 | 9.2 | 4.4 | 5.5 | 2.1 | 11.6 | - | - |
| MaskCLIP [ECCV22] | ✗ | 38.8 | 23.6 | 20.6 | 74.9 | 26.4 | 16.4 | 12.6 | 9.8 | 27.9 | - | - |
| SCLIP [ECCV24] | ✗ | 59.1 | 30.4 | 30.5 | 80.4 | 34.2 | 22.4 | 32.2 | 16.1 | 38.2 | 60 | 1580 |
| ClearCLIP [ECCV24] | ✗ | 51.8 | 32.6 | 33.0 | 80.9 | 35.9 | 23.9 | 30.0 | 16.7 | 38.1 | 45 | 650 |
| CdamCLIP [ICLR25] | ✗ | 58.7 | 30.6 | 35.2 | - | - | 24.8 | 23.7 | 17.2 | - | 100 | 920 |
| ResCLIP [CVPR25] | ✗ | 60.9 | 33.4 | 34.7 | 85.9 | 36.6 | 24.6 | 35.6 | 18.0 | 41.2 | 85 | 2550 |
| dino.txt [CVPR25] | ✗ | 35.9 | 22.9 | 24.2 | 84.5 | 26.1 | 19.2 | 22.6 | 18.9 | 31.8 | 90 | 1500 |
| FreeCP [ICCV25] | ✗ | 64.5 | 35.7 | 36.9 | 81.5 | 39.3 | 26.1 | 34.4 | 18.9 | 42.2 | 120 | 750 |
| SFP [ICCV25] | ✗ | 63.9 | 37.2 | 37.9 | 84.5 | 39.9 | 26.4 | 41.1 | 20.8 | 44.0 | 100 | 1600 |
| SC-CLIP [IEEE TIP25] | ✗ | 64.6 | 36.8 | 37.7 | 84.3 | 40.1 | 26.6 | 41.0 | 20.1 | 43.9 | 85 | 1600 |
| *VIP* (Ours) | ✗ | **73.2**↑8.6 | **41.2**↑4.0 | **47.4**↑9.5 | **92.5**↑6.6 | **46.5**↑6.4 | **33.3**↑6.7 | **55.7**↑14.6 | **29.1**↑8.3 | **52.4**↑8.4 | 100 | 1650 |

a single map during the final dense inference stage. We observe that naive aggregation ways, such as mean or max operations, fail to achieve optimal performance. This is mainly due to such schemes are vulnerable to outliers and noise, and tend to neglect the contextual information of the image. To address this, we propose the saliency-aware soft aggregation strategy to adaptively establish consensus regions among diverse linguistic prompts. We first compute the global response between the image global feature and distinct aliases within the same class, as follows:

$$\delta_k = \frac{\exp(g_k)}{\sum_{j=1}^{K} \exp(g_j)}, \text{ where } g = \text{cosine}(\sigma(I), \hat{T}_c), \quad (7)$$

where $\sigma(\cdot)$ is the global average pooling, and $\hat{T}_c \in \mathbb{R}^{K \times d}$ denotes the text embeddings of $K$ filtered aliases for the $c$-th category. We then utilize the global response $\delta \in \mathbb{R}^K$ as a saliency factor to modulate their dense activation maps:

$$\mathcal{S} = \delta \cdot \hat{\mathcal{M}}_c, \quad \text{where } \mathcal{S} \in \mathbb{R}^{hw \times K} \quad (8)$$

where $\hat{\mathcal{M}}_c \in \mathbb{R}^{hw \times K}$ represents the dense activation maps generated for the $K$ aliases via Eq. 1. Inspired by the energy-based model (LeCun et al., 2006; Liu et al., 2020), we leverage the free energy function to map multiple alias activations to a single dimension, formulated as:

$$\hat{\mathcal{S}} = \frac{1}{\tau} \log(\sum_k \exp(\tau \cdot \mathcal{S}_k)), \quad (9)$$

where $\tau$ is the temperature parameter. Unlike rigid max or mean operations, this probabilistic formulation mitigates spurious outliers and prevents signal dilution by rewarding the collective consensus of multiple aliases rather than relying on isolated or averaged activation maps.

### 3.5. Exploration of Transferability to Text Templates

In general, a complete text input comprises a class name and text templates. An example input prompt is "*a photo of the {class_name}*". Current studies are grounded in a set of 80 manually curated text templates offered by CLIP (Radford et al., 2021), originally tailored for image classification. Our expansion and filtering strategies (§3.2 and §3.3) can be equally transferable to the refinement of text templates. To be specific, we leverage LLM to generate candidate templates for dense prediction, and subsequently score these templates by fixing the filtered class names. Our attempt can be regarded as a preliminary exploration of text templates and demonstrates the transferability of the proposed method. Additional details are provided in the Appendix A.1.

## 4. Experiments

### 4.1. Experimental Setup

**Datasets.** We conduct our main experiments on eight gold-standard benchmarks: *i) with a background class*: PAS-

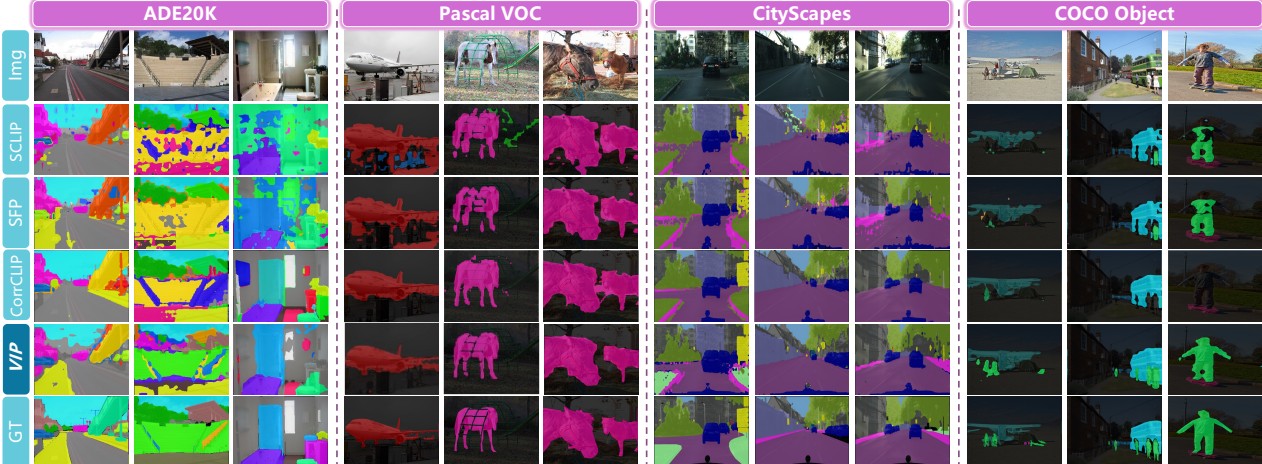

*Figure 4.* **Qualitative comparison** results on natural images and urban scenes. Additional results are provided in the Appendix B.3.

*Table 2.* **Quantitative comparison** on remote sensing imagery. Our *VIP* surpasses current SOTA methods by a substantial margin, including SegEarth-OV, which is tailored for remote sensing.

| Methods | iSAID | Vaihing. | Potsdam | VDD | Avg. |
|---|---|---|---|---|---|
| CLIP [ICML21] | 7.5 | 10.3 | 14.5 | 14.2 | 11.6 |
| ProxyCLIP [ECCV24] | 20.7 | 27.8 | 44.1 | 44.3 | 34.2 |
| dino.txt [CVPR25] | 19.3 | 18.9 | 24.3 | 32.3 | 23.7 |
| Trident [ICCV25] | 20.0 | 27.7 | 44.4 | 45.7 | 34.5 |
| CorrCLIP [ICCV25] | 16.9 | 24.7 | 42.6 | 37.7 | 30.5 |
| SC-CLIP [IEEE TIP25] | 18.4 | 29.6 | 43.4 | 41.0 | 33.1 |
| SegEarth-OV [CVPR25] | 21.7 | 29.1 | 47.1 | 45.3 | 35.8 |
| *VIP* (Ours) | 26.1 ↑4.4 | 47.0 ↑17.4 | 49.8 ↑2.7 | 54.3 ↑8.6 | 44.3 ↑8.5 |

CAL VOC (Everingham et al., 2015) (**VOC21**), PASCAL Context (Mottaghi et al., 2014) (**Context60**), COCO Object (Caesar et al., 2018) (**Object**); and *ii) without background class*: PASCAL VOC (Everingham et al., 2015) (**VOC20**), PASCAL Context59 (Mottaghi et al., 2014) (**Context59**), COCO Stuff (Caesar et al., 2018) (**Stuff**), Cityscapes (Cordts et al., 2016) (**City**) and ADE20K (Zhou et al., 2019) (**ADE**). To assess the generalization capability of our approach across different domains, we additionally perform evaluation on four widely used remote sensing semantic segmentation benchmarks, *i.e.*, **iSAID** (Waqas Zamir et al., 2019), **Potsdam**[1], **Vaihingen**[2], and **VDD** (Cai et al., 2025).

**Evaluation Metric.** We employ the standard mean Intersection over Union (mIoU) as the evaluation metric. All results are not post-processed with mask-refinement

[1] https://www.isprs.org/resources/datasets/benchmarks/UrbanSemLab/2d-sem-label-potsdam.aspx

[2] https://www.isprs.org/resources/datasets/benchmarks/UrbanSemLab/2d-sem-label-vaihingen.aspx

methods like DenseCRF (Krähenbühl & Koltun, 2011) and PAMR (Araslanov & Roth, 2020) for a fair comparison. We employ the average per-image inference time and memory footprint as evaluation metrics for computational overhead.

**Implementation Details.** We adopt DINOv3 (Siméoni et al., 2025) with ViT-L (Dosovitskiy et al., 2021) as the visual backbone for dense inference. For each class name, we query GPT-5 by default to generate candidate aliases and text templates. Our implementation is based on mmsegmentation repository, and all experiments are conducted on a NVIDIA RTX 4090 GPU. We empirically set the hyperparameters guided by prior knowledge of the dataset's semantic complexity and class granularity.

### 4.2. Comparison to State-of-the-Arts

**Counterparts.** We categorize existing CLIP-based methods into two groups: ① requiring auxiliary visual foundation models (*dense prediction with multi vision models*), and ② relying exclusively on CLIP (*dense prediction with single vision model*). As *VIP* does not rely on any auxiliary vision models, we classify it into the second group. We provide additional details about the counterparts in Appendix A.2.

**Quantitative Comparisons on Natural Images and Urban Scenes.** Table 1 reports the quantitative comparison of our method against existing training-free OVSS approaches. In particular, for single model competitors, *VIP* surpasses the recent top-leading method SFP (Jin et al., 2025) by a large margin, yielding a remarkable gain of **8.4%** average mIoU. Even against the multi-model SOTA CorrCLIP (Zhang et al., 2025), which leverages auxiliary DINO (Caron et al., 2021) and SAM2 (Ravi et al., 2025) to counter the spatial bias of CLIP, our method still exhibits superior performance, achieving **1.4%** average improvements. Crucially, this is accomplished while delivering up to **14×**

*Table 3.* **Ablation experiments** of key components. For simplicity, we use the abbreviations of the modules. The max operation is applied to aggregate activation maps of aliases for baseline strategy.

| Methods | VOC | Object | City | ADE | Avg. | Δ |
|---|---|---|---|---|---|---|
| baseline | 35.9 | 24.2 | 22.6 | 18.9 | 25.4 | - |
| + *S-C* §3.1 | 62.1 | 31.4 | 35.0 | 23.6 | 38.0 | - |
| + *SE-LLM* §3.2 | 61.9 | 28.6 | 36.7 | 23.0 | 37.6 | -0.4 |
| + *Alias Dis.* §3.3 | 68.3 | 41.3 | 50.2 | 27.1 | 46.7 | +8.7 |
| + *Soft Aggr.* §3.4 | 72.3 | 46.7 | 54.8 | 28.9 | 50.7 | +12.7 |
| + *Templates.* §3.5 | **73.2** | **47.3** | **55.7** | **29.1** | **51.3** | +13.3 |

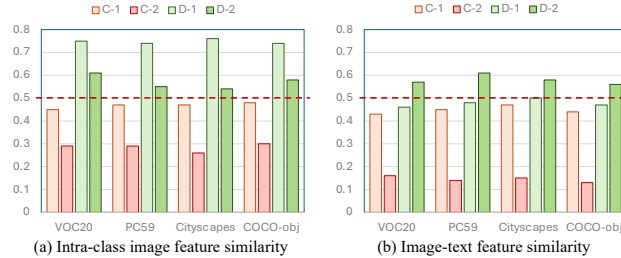

(a) Intra-class image feature similarity  (b) Image-text feature similarity

*Figure 5.* **Analysis experiments of spatial bias.** Here 'C' and 'D' denote the CLIP and dino.txt model respectively, while the '-1' and '-2' represent the modulated layer numbers.

faster inference speed and halving the memory footprint. *In light of the potential that* **manual mask annotations** *from these public datasets are partially or entirely incorporated into the training data for SAM2, our results are particularly compelling.*

**Quantitative Comparisons on Remote Sensing Images.** Table 2 shows our superior performance on remote sensing images. It is observed that most CLIP-based solutions fail to generalize effectively to out-of-distribution domain. In contrast, our *VIP* remarkably surpasses the existing approaches, demonstrating strong generalization ability. Although SegEarth-OV (Li et al., 2025a) capitalizes on pre-trained upsampling components tailored for remote sensing, *VIP* transcends this domain-specific barrier with a significant **8.5%** performance boost in average mIoU. These findings confirm that *VIP* unlocks the *generic* fine-grained perception potential of dino.txt, demonstrating its robust generalization across diverse scenarios.

**Qualitative Results.** Figure 4 provides qualitative comparisons of *VIP* against prior CLIP-based solutions. We observe that while CLIP-based SOTA methods can capture salient foreground regions, they are prone to the pitfall of semantic distortion, like 'grandstand', 'pave-way' on the ADE20K and 'terrain' on the CityScapes. In contrast, our *VIP* effectively bridges this gap, generating high-fidelity masks while maintaining superior semantic understanding.

### 4.3. Diagnostic Experiment

**Key Component Analysis.** To better understand the role of each component in our *VIP*, we conduct an ablation study across four datasets based on dino.txt. As shown in Table 3, dino.txt is capable of rectifying the spatial bias induced during global pre-training (we provide further analysis below), thereby enhancing segmentation performance. Yet, the results are still unsatisfactory, lagging behind several CLIP-based SOTA. The $4^{th}$-$6^{th}$ rows indicate that our proposed components are able to work in a collaborative manner, effectively alleviating the dense cross-modal mismatch in dino.txt and yielding marked performance gains. It is worth noting that directly utilizing LLM-generated aliases without

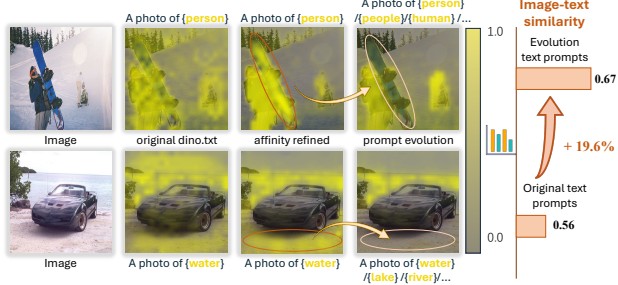

*Figure 6.* **Analysis experiments of text query responses.** Activation values are normalized to the range [0, 1]. The erroneous responses highlighted are effectively corrected through our *VIP*.

filtering has a detrimental impact, as this introduces excessive noise. In addition, the efficacy of our core idea on text templates ($6^{th}$ row) highlights its exceptional transferability and opens an avenue for future in-depth exploration.

**dino.txt versus CLIP.** To investigate the pervasiveness of spatial bias within CLIP and dino.txt, we conduct quantitative empirical analyses. Specifically, we consider two comparative settings for both models: ① modulating the self-attention mechanism exclusively in the final layer of the image encoders, and ② extending the modulation to the penultimate layer to further enhance spatial awareness. Figure 5 shows the intra-class image feature similarity and image-text feature similarity under the two settings. It is evident that CLIP's cross-modal alignment suffers a collapse following extended modulation This reveals that the spatial bias in CLIP is inextricably entangled with its cross-modal alignment capability. Hence, attempts to address the former often come at the expense of the latter. Conversely, benefiting from the inherent spatial awareness priors of DINOv3, dino.txt fundamentally mitigates this bottleneck. More detailed analyses are provided in the Appendix B.2.

**Comparative Study on Text Query Activations.** We further conduct a comparative study of the activation quality induced by original text query versus refined text queries on dense image features. As shown in Figure 6, while the self-correction mechanism suppresses artifacts for original

queries, significant noises remain persistent. In contrast, the refined text queries exhibit more holistic and pristine activations, remarkably eliminating noises and accurately delineating object regions. To quantify the impact of refined text queries on cross-modal interaction, we computed the similarity scores between them and the dense image features. Our method delivers a $19.6\%$ increase in image-text similarity, narrowing the gap between dense image features and corresponding semantic anchors, thereby mitigating the cross-modal mismatch limitation.

## 5. Conclusion

In this paper, we depart from the prevailing CLIP-based paradigm and exploit the untapped potential of dino.txt to achieve efficient and high-quality training-free OVSS. Towards this end, we introduce **VI**sual-guided **P**rompt evolution (*VIP*), to mitigate the dense cross-modal mismatch in dino.txt, thereby unleashing its potent object perception capabilities. Experimental results validate that *VIP* remarkably outperforms existing top-leading methods on diverse challenging domains, while incurring only negligible inference costs. These results are non-trivial, especially considering that our approach relies solely on self-supervised pre-trained foundation model, without recourse to any potential mask supervision. We hope that this work can inspire further innovation towards this promising avenue.

## Acknowledgments

This work is supported by National Key R&D Program of China (2023YFD2000303) and National Natural Science Foundation of China (62372433).

## Impact Statement

This paper presents work whose goal is to advance the field of machine learning. There are many potential societal consequences of our work, none of which we feel must be specifically highlighted here.

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

The appendix is divided into several sections, each giving extra information and details.

## A. Detailed Experimental Setups

### A.1. Implementation Details

**Main Experiment.** The original version of dino.txt (Jose et al., 2025) adopted DINOv2 (Oquab et al., 2024) as its visual backbone, and was then upgraded in tandem with the release of DINO v3 (Siméoni et al., 2025). In this work, we employ the latest version to ensure optimal performance. To ensure a fair comparison, our data processing pipeline strictly follows to the protocols established in previous studies (Wang et al., 2024; Zhu et al., 2024; Chi et al., 2025; Kang et al., 2025). Specifically, we first resize each input image with the shorter side fixed at 336 pixels (or 560 pixels for the high-resolution Cityscapes dataset), followed by sliding-window inference using a window size of 224 and a stride of 112. Furthermore, during the aggregation phase, we incorporate the activation maps derived by canonical class name to ensure semantic stability. For hyperparameter settings, we empirically observe that the propagation converges when $\alpha$ and $\beta$ are set to 2 (*cf.* Eq.4). Thus, we adopt this setting as default to facilitate efficient propagation. The temperature parameter $\tau$ (*cf.* Eq.9) is set to 4.0. The comparison of the framework pipeline between the CLIP-based paradigm and our *VIP* is shown in Figure 7.

**Measurement of Inference Cost.** In general, the semantic expansion (§3.2) and alias distillation (§3.3) are conducted offline, whereas the final aggregation (§3.4) is performed online to yield the final segmentation results, as illustrated in Figure 3. However, in practice, the alias distillation process already computes the similarity maps between the dense image features and all text queries, which can be cached. This enables the aggregation of logits maps and the final segmentation results can be obtained directly, bypassing the need for a repeated model forward pass, thereby significantly reducing inference overhead. **The average inference speed and memory footprint reported in Table 1 are calculated in this way to assess the true cost of our method in practical deployment.** Notably, our alias distillation and aggregation modules rely exclusively on a few straightforward matrix multiplications, incurring negligible latency and achieving an optimal trade-off between performance and inference costs.

**Prompting the Language Model.** A key component of our method involves leveraging LLM (we use GPT-5 by default) to generate image-caption style aliases for each category, thereby expanding semantic coverage. Recall that the form of our prompt introduced in §3.2:

---

`Query:` Generate 20 image-caption style noun phrases for {category name}, covering synonyms, plurals, hyponyms, and concrete visual variants.

---

In practice, we typically incorporate specific scene descriptors into the prompt to guide the LLM in generating scene-specific class aliases. Consistent with previous work utilizing GPT (Menon & Vondrick, 2023; Pratt et al., 2023), we provide several examples of the expected output to enhance the reliability of list formatting. These examples may take the following form:

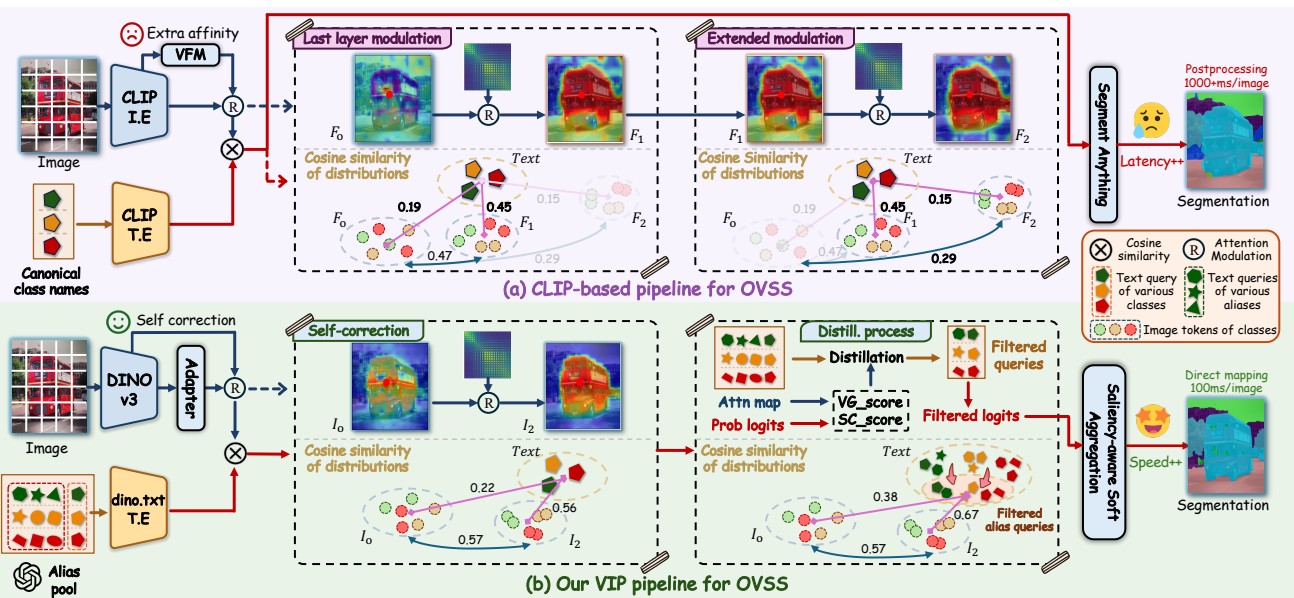

*Figure 7.* **Framework comparison of CLIP-based paradigm and our *VIP* pipeline.** Beneath the module diagram, we illustrate the flow of image-text similarity distributions. Here, $F$ and $I$ denote the dense image features of CLIP-based paradigm and dino.txt, respectively, with subscripts indicating the number of attention modulation iterations the features have undergone.

---

**Query:** Generate 20 image-caption style noun phrases for {bus} in {natural images}, covering synonyms, plurals, hyponyms, and concrete visual variants.
**Answer:** city bus, buses, double-decker bus, coach bus, minibus, tour bus, passenger bus, articulated bus, shuttle bus, transit bus, electric bus, vintage bus, luxury bus, yellow bus, red bus, open-top bus, night bus, intercity bus, express bus, blue bus

---

To mitigate potential hallucinatory outputs, we discard aliases exhibiting a cosine similarity lower than $0.7$ with the canonical class name by default. It is possible that extensive prompt tuning would produce better aliases; however, this simple formulation proves sufficient. Since our goal is simple, the critical challenge is the filtration of these aliases to retain useful information, which constitutes the core of §3.3.

**Attempt on Transferability to Text Templates.** Our core idea applied to class names can be similarly transferred to text templates. Currently, training-free OVSS methods predominantly rely on the 80 hand-crafted templates provided by CLIP, which are specific to image classification. In fact, these templates encompass a vast majority of application scenarios and are consequently widely adopted by existing zero-shot visual understanding tasks. Therefore, we utilize these templates as references for LLM to generate new text templates tailored for the segmentation task across diverse scenarios. We upload the CLIP's 80 text templates as an attachment to LLM in advance, and subsequently employ prompts in the following format to query LLM:

---

**Query:** Generate 50 diverse CLIP-style text templates tailored for segmentation task. The generated templates should capture the specific visual signatures and contextual semantics of {natural images}.

---

For each generated text template, we compute the corresponding VG_Score (*cf.* Eq. 5) and SC_Score(*cf.* Eq. 6), and utilize the mean scores of the 80 CLIP templates as a reference. Mirroring our filtering strategy for class aliases, we retain only those templates that surpass the reference value in VG_Score while remaining lower than the reference in SC_Score. In addition, we maintain two foundational templates across all scenarios, *i.e.*, "*a photo of a* {*class_name*}" and "*a detailed view of a* {*class_name*}", to ensure a stable semantic representation. Though somewhat modest compared to those for class names, the achieved gains on text templates are still sufficient to demonstrate the generalizability of our idea. Given that it is not the primary focus of our work, we encourage future research to conduct a more in-depth exploration of text templates.

**Implementation Details of Spatial Bias Experiments.** CLIP-based methods typically modify the last layer of the image encoder to enhance the feature affinity for improved segmentation results. However, only one-layer refinement often exhibits sub-optimal and flawed dense image features. Hence, we extend this refinement operation to the penultimate layer for

diagnostic analysis, which validates how image feature affinity influences the cross-modal alignment representation. To quantitatively analyze these variant models, we calculate intra-class image feature similarity and image-text feature similarity scores, respectively. The results are summarized in Figure 8 and the detailed calculation processes are as follows:

**I)** Image features. We first use ground-truth (GT) masks to isolate patch tokens for each class and accordingly separate them into different classes. For patch tokens refined at the last layer and those further refined at the penultimate layer, we compute the cosine similarity between each refined patch token and its corresponding original token in a per-patch manner. The mean of these similarities is finally reported as the intra-class image-feature similarity score. In this way, a higher score indicates that the refined patch tokens remain closer to the original feature distribution, whereas a lower score suggests greater deviation from it.

**II)** Cross-modal features. Similarly, we first utilize GT masks to isolate patch tokens for each class and group them accordingly. Then, we compute the cosine similarity between each patch token and the corresponding class-specific text embedding (*e.g.*, "*a photo of the {class_name}*"). Finally, we average the similarity scores over all patch tokens within each class to obtain the image–text feature similarity score for that class. Under this formulation, a higher score indicates stronger discriminative alignment between the patch tokens and the corresponding class, and vice versa.

### A.2. Counterparts

We categorize existing training-free OVSS CLIP-based solutions into two groups based on whether they rely on auxiliary visual foundation backbones during dense inference, as follows:

**I)** *dense prediction with multi vision models*: LaVG (Kang & Cho, 2024), ProxyCLIP (Lan et al., 2024b), LPOSS (Stojnić et al., 2025), CASS (Kim et al., 2025), FSA (Chi et al., 2025), CLIPer (Sun et al., 2025), Trident (Shi et al., 2025), CorrCLIP (Zhang et al., 2025).

**II)** *dense prediction with single vision model*: CLIP (Radford et al., 2021), MaskCLIP (Zhou et al., 2022), SCLIP (Wang et al., 2024), ClearCLIP (Lan et al., 2024a), CdamCLIP (Kang et al., 2025), ResCLIP (Yang et al., 2025), FreeCP (Chen et al., 2025), SFP (Jin et al., 2025), SC-CLIP (Bai et al., 2025). As our *VIP* does not require auxiliary visual models for dense inference, it is categorized within this group. Furthermore, we observe that the majority of these methods perform better on ViT-B than on ViT-L backbone. This is primarily attributed to the fact that increased network depth exacerbates the spatial bias encoded in CLIP, thereby leading to inferior performance. For this reason, we report the better performance (using ViT-B model) of these methods to ensure a fair comparison.

## B. Additional Experimental results

### B.1. Diagnostic Analysis

**Effect of Different Language Models.** We evaluate the impact of using different LLMs for alias generation. Specifically, we queried Gemini-2.5 (Comanici et al., 2025) and DeepSeek-R1 (Guo et al., 2025) using the identical prompt, with the results reported in Table 4. It can be seen that the choice of LLM has a negligible impact on performance, and we find that the category aliases generated by different models are largely consistent. This is primarily because alias generation is a straightforward task for modern LLMs; given the prompt and demonstration examples, all models can perform it well.

*Table 4.* **Ablation studies of different large language models.** Broadly speaking, the influence of utilizing distinct LLMs is minimal, attributed to the fact that all models are adept at the task of generating semantically diverse aliases.

| *LLMs* | VOC21 | Object | City | ADE | **Avg.** |
|---|---|---|---|---|---|
| *Gemini-2.5* (Comanici et al., 2025) | 73.0 | 47.2 | 55.4 | 28.9 | 51.1 |
| *DeepSeek-R1* (Guo et al., 2025) | 72.8 | 46.5 | 55.1 | 28.8 | 50.8 |
| *GPT-5* | 73.2 | 47.3 | 55.7 | 29.1 | 51.3 |

**Semantic Expansion by Category Descriptors.** Leveraging LLMs to generate visual context descriptors for each category has been extensively studied in prior works on zero-shot image classification via VLMs (Menon & Vondrick, 2023; Pratt et al., 2023; Roth et al., 2023). We also adopted this practice in our early experimental validation. However, we found it unsuitable for segmentation task. In particular, we directly employ the prompts provided in prior works to query GPT-5, generating corresponding context descriptors for each category. The formats of the prompt are as follows (please refer to CuPL (Pratt et al., 2023) for details):

*Table 5.* **The performance of category descriptor queries.** Deriving text queries via category descriptors leads to a severe performance drop. Notably, this issue persists even after applying our filtering mechanism.

| Methods | VOC21 | Object | City | ADE | **Avg.** | Δ |
|---|---|---|---|---|---|---|
| baseline | 35.9 | 24.2 | 22.6 | 18.9 | 25.4 | - |
| + *S-C* §3.1 | 62.1 | 31.4 | 35.0 | 23.6 | 38.0 | - |
| + *Category Descriptor* | 58.2 | 27.6 | 32.8 | 21.0 | 34.9 | -3.1 |
| + *Descriptor Distillation* | 59.3 | 27.4 | 33.2 | 21.5 | 35.4 | -2.6 |

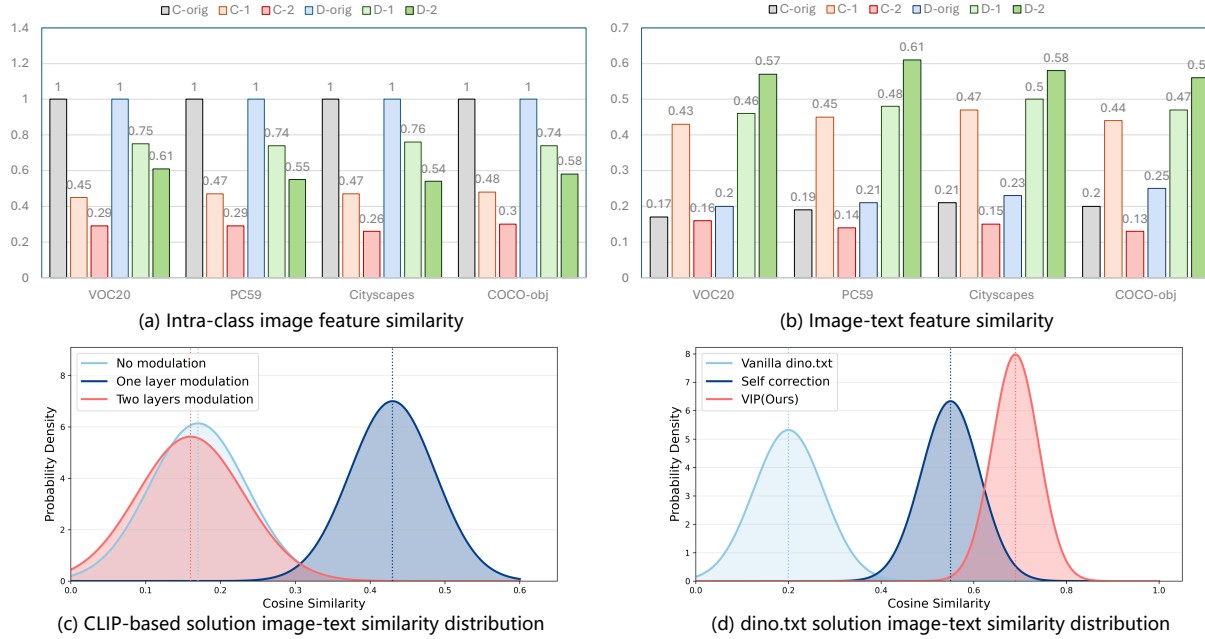

*Figure 8.* **Quantitative similarity analysis** between the refined and the original features. 'C' and 'D' denote the CLIP and dino.txt models, respectively, while the '-1' and '-2' denote the refined layers. '-orig' denotes the original feature. (a) Intra-class image feature similarity, which measures the patch-level similarity between the refined and the original image features. '-orig' means the self-similarity, *e.g.*, $= 1$. (b) Image-text similarity, which measures the image-text similarity among the refined and the original image features to the same text embeddings. (c) CLIP-based solution image-text similarity distribution. (d) dino.txt solution image-text similarity distribution. Both (c) and (d) represent results evaluated on the Pascal VOC.

> **Query:** (1) Describe what a(n) {} looks like, (2) How can you identify a(n) {}?, (3) What does a(n) {} look like?, (4) A caption of an image of a(n) {}, (5) Describe an image from the internet of a(n) {}.

We can obtain the following sample outputs for {bus} category:

> **Answer:** A bus is a large road vehicle designed to carry many passengers. A bus is a long, rectangular vehicle with multiple windows along its sides. A bus is a motor vehicle that typically has two or more axles and a high roof. A bus is a public or private transport vehicle commonly used in cities, schools, and for long-distance travel.

However, substituting the standard combination of templates and class names with such descriptors leads to a significant performance degradation, as shown in Table 5. We attribute this to the fact that dense image features primarily encode local visual semantics and lack the capacity to capture global contextual information. By contrast, in image classification task, where global image features are matched with text queries, the category descriptors can effectively enhance cross-modal perception. Consequently, unlike existing methods, we focus on a coarse-grained approach that refines aliases for segmentation, which offers a simple yet effective way to unleash the dense perception ability of dino.txt.

### B.2. More Experimental Analysis

**More Analyses on dino.txt versus CLIP.** As discussed in main paper, the image feature distributions of dino.txt exhibit greater stability than CLIP, which avoids modality alignment collapse. To substantiate this claim, we apply affinity

*Table 6.* **The precision of semantic affinity in the dense image features**. Results are evaluated on Pascal VOC dataset.

| Methods | SCLIP (Wang et al., 2024) | ProxyCLIP (Lan et al., 2024b) | SC-CLIP (Bai et al., 2025) |
|---|---|---|---|
| *One layer modulation* | 46.4 | 61.5 | 60.8 |
| *Two layers modulation* | 54.7 | 73.8 | 69.6 |
| $\Delta$ | +8.3 | +12.3 | +8.8 |

refinement to the last-layer image features of both dino.txt and CLIP, and further extend this operation to the penultimate layer. Build upon this, we conduct a quantitative analysis of the induced changes, which assesses their variations in intra-class image-feature similarity between the refined feature and the original feature. First, we demonstrate the impact of extended modulation on the semantic affinity of image features in Table 6 (refer to the preceding text for implementation details). It is evident that two-layer modulation significantly enhances the semantic affinity of features, rendering them more discriminative. Subsequently, Figure 8 (a) shows that the extended affinity refinement from an additional VFM leads to a substantial drop in CLIP's intra-class image-feature similarity, *e.g.* $1 \to 0.3$. In contrast, the reduction for dino.txt is much lower and limited to 50%. This discrepancy arises because the external VFM and CLIP exhibit mismatched feature distributions, whereas dino.txt relies on features from its own frozen backbone, thereby preserving the original feature distribution.

We also quantitatively evaluate the induced changes in image-text alignment, shown in Figure 8 (b). It illustrates that both the original CLIP and dino.txt exhibit unpromising image-text similarity, which suggests that if we do not modify their original vision transformer process, they show a poor segmentation performance on OVSS. After the affinity refinement is applied, both CLIP and dino.txt achieve a significant improvement ($0.18 \to 0.45$ for CLIP, and $0.23 \to 0.49$ for dino.txt). Moreover, when the extended refinement is employed, CLIP suffers a catastrophic degradation, *e.g.* $0.45 \to 0.15$. It indicates that the extended refinement on CLIP disrupts the unified feature space from the pre-training stage. In contrast, dino.txt performs self-affinity refinement derived from its backbone features, consistently improving the image–text alignment, *e.g.* $0.46 \to 0.57$. These results demonstrate a robust unified feature space of dino.txt for cross-modal inference and highlight the inherent image-text limitation of CLIP-based paradigms in spatial optimization, as elaborated in §3.1.

**B.3. More Qualitative Segmentation Results**

In Figure 9, we present qualitative comparisons across four representative remote sensing semantic segmentation benchmarks. As observed, our *VIP* exhibits superior semantic comprehension while precisely delineating object boundaries, thereby substantially surpassing prior state-of-the-art methods.

We also provide more qualitative comparisons between our *VIP* and existing counterparts, including Trident (Shi et al., 2025), SFP (Jin et al., 2025), ResCLIP (Yang et al., 2025), SCLIP (Wang et al., 2024), and the baseline dino.txt (Jose et al., 2025) on **Object**, **ADE**, **Context60**, and **VOC21** benchmarks, respectively. As shown in Figure 10-13, through the proposed text evolution, our *VIP* successfully corrects misclassifications and avoids missing objects found in existing counterparts.

## C. Limitations and Future Work

### C.1. Limitations

Although VIP can substantially improve the quality of text queries in dino.txt, the model may still fail in certain edge cases. For example, in the COCO-Stuff dataset, the definitions of some category names are highly ambiguous, such as *wall-concrete* and *wall-panel*. While VIP enhances the model's semantic representation ability for most categories, its segmentation performance remains poor on these highly confusable classes. In addition, although VIP can filter category names generated by LLMs, the selected category aliases are not necessarily optimal due to the lack of explicit supervision signals. There is therefore still considerable room for improvement.

### C.2. Future Work

Our work reveals that, under the current paradigm, the main bottleneck in training-free OVSS with VLMs lies not in obtaining more discriminative image features, but in enhancing the model's semantic recognition ability. Future work could therefore focus on improving the expressive power of the text branch, for example by mining better text templates or precisely correcting category names.

In addition, in terms of generalization, our model is limited by the DINOv3 pretrained backbone and may perform poorly on low-resolution images, such as Sentinel-2 satellite imagery (Zhu et al., 2025). Future work could further explore how to generalize the model's capability in this direction.

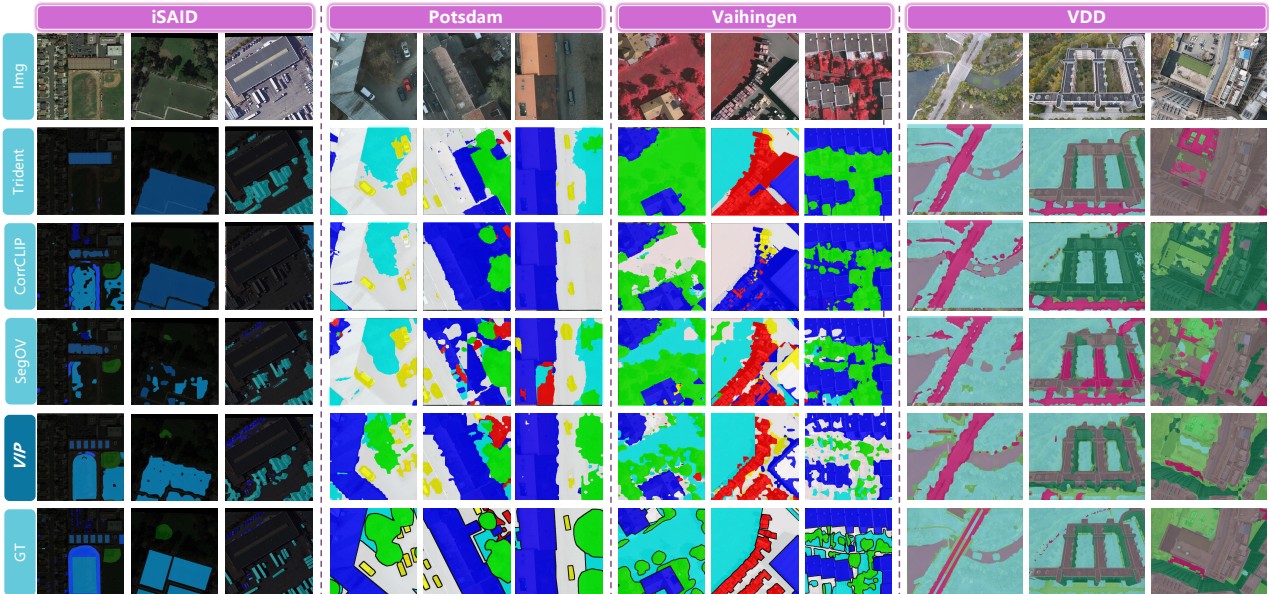

*Figure 9.* **Qualitative comparison** results on remote sensing imagery.

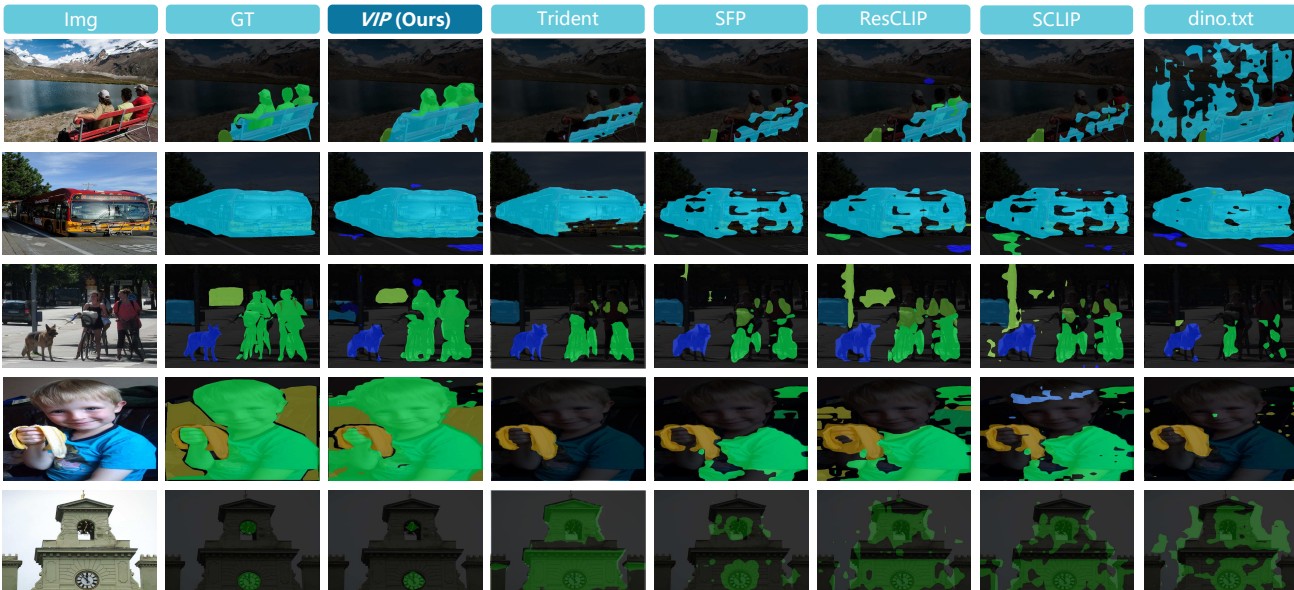

*Figure 10.* Qualitative comparison on the Object benchmark.

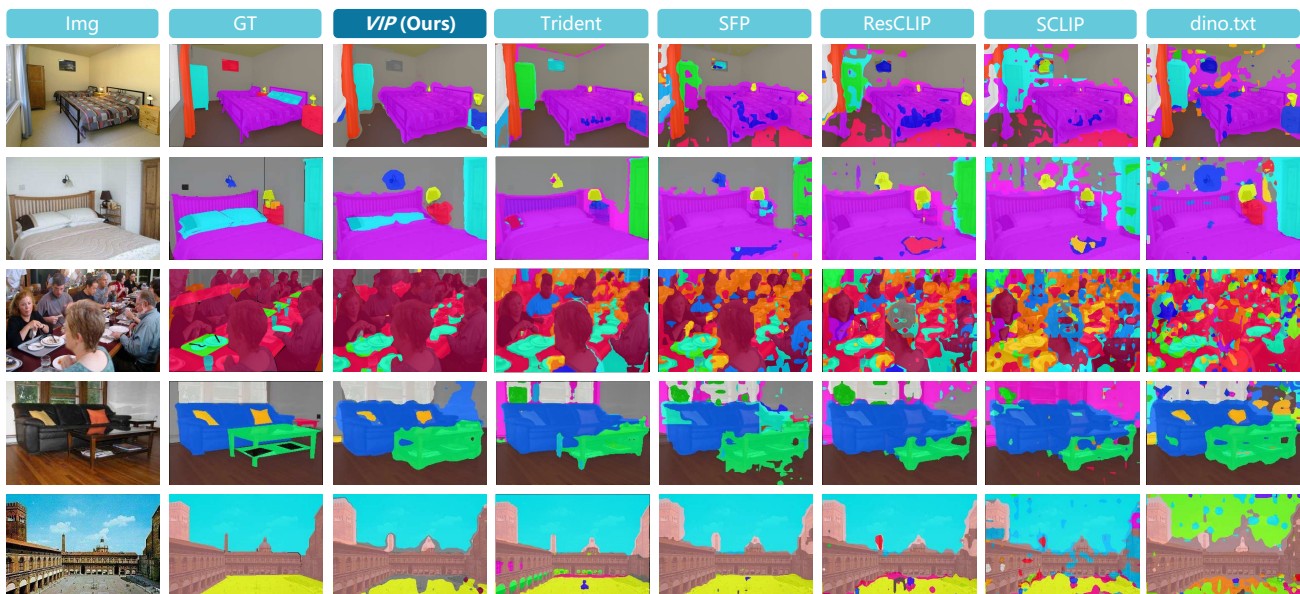

*Figure 11.* Qualitative comparison on the ADE benchmark.

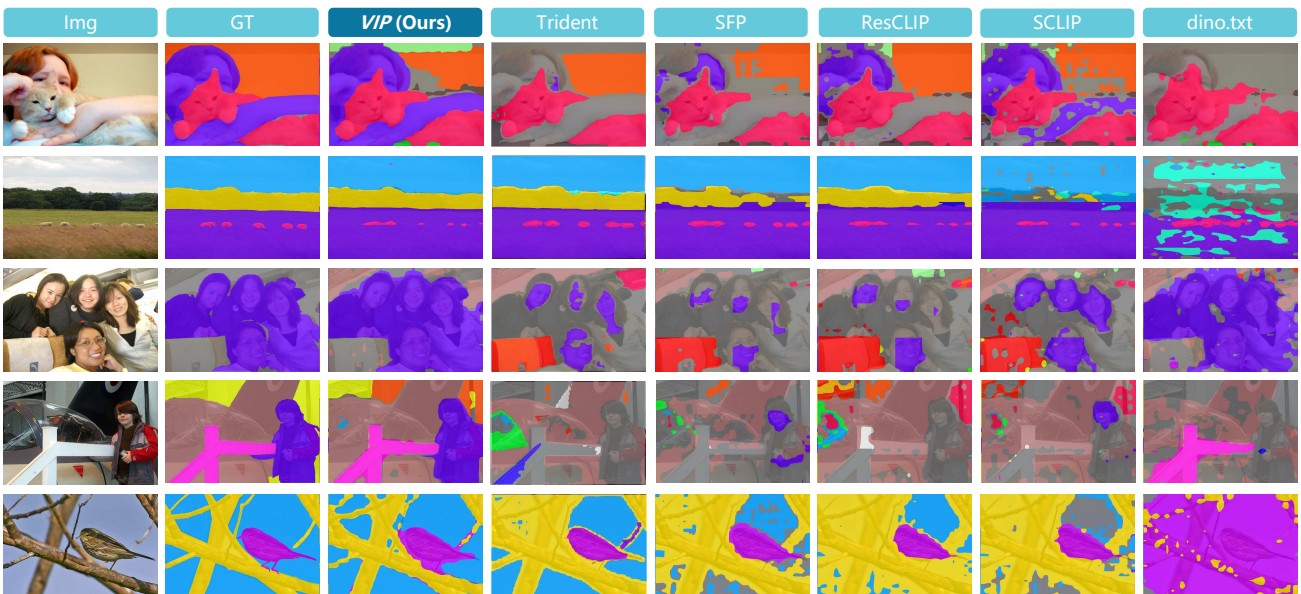

*Figure 12.* Qualitative comparison on the Context60 benchmark.

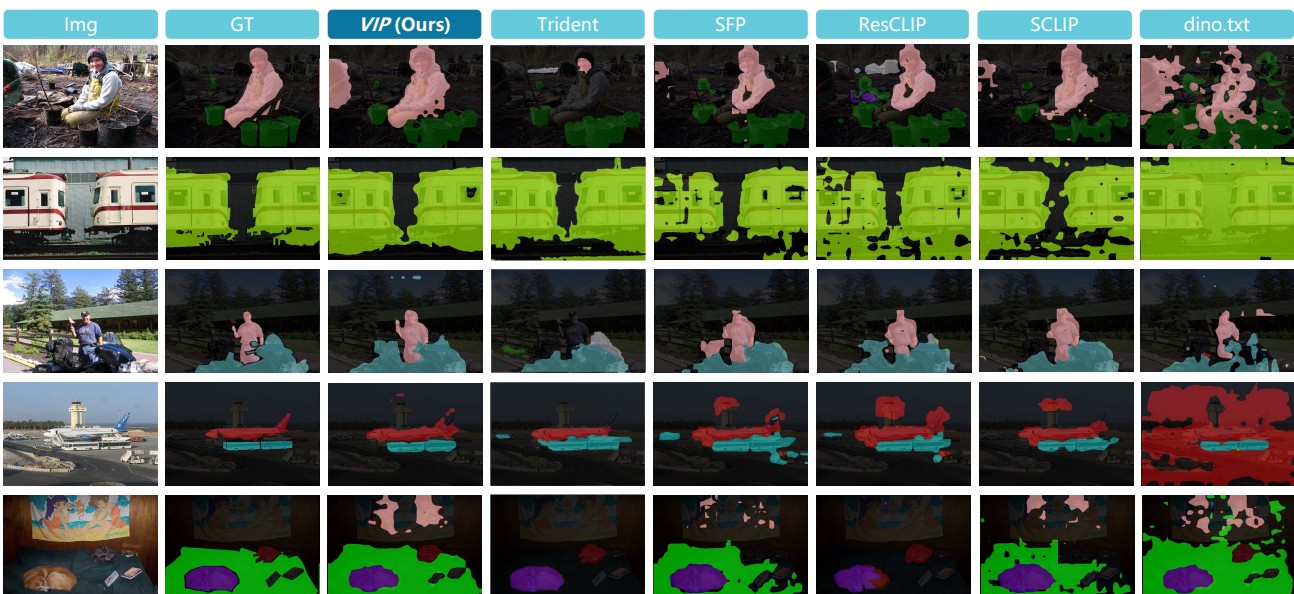

*Figure 13.* Qualitative comparison on the VOC21 benchmark.

