# OpenReview forum: "VIP: Visual-guided Prompt Evolution for Efficient Dense Vision-Language Inference"
_ICML.cc/2026/Conference — ICML 2026 regular_

### Official Review · Reviewer_KtmB · 2026-02-25

**Soundness:** 3
**Presentation:** 3
**Significance:** 3
**Originality:** 3
**Overall Recommendation:** 4
**Confidence:** 4

**Summary:**

The paper introduces Visual-guided Prompt Evolution (VIP). The main idea is to conduct semantic enrichment for text queries, with guidance from visual information, which can greatly improve how images and text match with each other. VIP has three main parts. First, it uses a large language model to expand class names into simple caption-style names. Second, it selects and filters these names with visual guidance, using DINO attention and simple scoring methods (VG and SC scores). Third, it combines the results from different names in a way that focuses more on important image areas. The method does not need training and does not depend on extra segmentation models like SAM.
Extensive results across various benchmarks demonstrate consistent improvements over prior single-model and multi-model training-free OVSS approaches, while maintaining competitive inference efficiency. Ablation and diagnostic analyses further support the proposed design choices.

**Compliance With Llm Reviewing Policy:**

Affirmed.

**Key Questions For Authors:**

1. Could the authors provide a performance sensitivity analysis when applying different LLMs for semantic enrichment?
2. Could the authors provide some failure case analysis to help gain a better understanding of VIP?
3. What is the computational cost of alias distillation when scaling to hundreds or thousands of categories? Does the dataset-wide scoring become a bottleneck?
4. If alias filtering is done on one dataset (e.g., VOC), can the same alias library transfer effectively to another dataset with overlapping classes?

**Limitations:**

yes

**Strengths And Weaknesses:**

## Strengths

1. The technical formulation is coherent and logically motivated. The identification of cross-modal mismatch in dino.txt, rather than purely spatial bias, is well argued. Especially, the diagnostic experiments comparing CLIP and dino.txt under different attention modulations are informative and support the claim that spatial bias is more tightly coupled with cross-modal alignment in CLIP.
2. Evaluation is broad, including eight natural/urban benchmarks and four remote sensing datasets, which strengthens generalization claims.

## Weakness

1. The LLM-based alias generation uses GPT-5 by default, but the performance sensitivity to the choice of LLM is not deeply analyzed.
2. Despite the paper's claim as a training-free method, the alias library is optimized for the evaluation dataset. Performance might drop if deployed in an unseen environment where dataset-wide passes are not possible.
3. Lack of failure case analysis. Where does VIP fail? Does alias filtering collapse for visually ambiguous classes? What happens when classes are extremely fine-grained? A deeper failure analysis would strengthen the paper.

---

> ### Author Rebuttal · Authors · 2026-03-31
>
> > **`W1&Q1: LLM sensitivity`**
>
> We apologize that this point was not sufficiently clarified in the main paper. Due to space limits, our analysis of LLM sensitivity was provided in **Appendix C.1 (Table 4)** of submitted manuscript, which reveals that VIP is not sensitive to the specific LLM choice and works well across all models. We also provide more results (e.g., on lighter open-source LLMs) and discussion on this part, please see our response to **`#Vvdx-W3`**.
>
> ----
>
> > **`W2: Dataset-wide process`**
>
> We appreciate the reviewer for raising this important point! To address your concern, we would like to highlight that:
>
> - *VIP* is more akin to a **self-correction** process rather than a dataset-level optimization. As noted in **Appendix B.1 (Line 855-864)**, we can obtain the dense image-text segment maps for all aliases during the distillation, thus, the entire pipeline requires **only one single forward pass for each dataset**. Moreover, the whole process strictly follows the *training-free* paradigm, and is consistent with common practice for evaluating training-free methods on batches of target images.
>
> - We further evaluate a single-image variant (**Table-Single** ([`Anonymous link`](https://anonymous.4open.science/r/ICML_2320/Table-Single.md))), where alias filtering is conducted independently for each image and then applied to the same image. It still **substantially outperforms the canonical baseline** (by 10.3 average mIoU), indicating that our approach remains highly effective even in a single-image setting.
>
> - We also conduct a cross-split transfer study (**Table-Split** ([`Anonymous link`](https://anonymous.4open.science/r/ICML_2320/Table-Split.md))): the alias library is distilled on the training split and applied directly to the test split without any re-scoring. This setting also works well, and on most datasets even surpasses the version built from the test split. This shows that VIP can leverage a pre-collection of unlabeled samples to build an alias library that generalizes well to unseen data. More broadly, it highlights VIP is **data-scalable**: as more unlabeled data become available, alias distillation becomes more accurate.
>
> -----
>
> > **`W3&Q2: Failure case analysis`**
>
> We greatly appreciate your constructive suggestion! We provide failure cases and descriptions in the **Fig-Failure** ([`Anonymous link`](https://anonymous.4open.science/r/ICML_2320/Fig-Failure.pdf)). Our analysis reveals two primary failure modes of VIP:
>
> - **Over-sensitivity to specific visual cues.** This appears to be a side effect of improved semantic coverage. For instance, when "wall" is expanded to "stone wall," the model becomes highly sensitive to stone-related contexts, which can cause false positives on visually similar objects such as '*building*' or '*rock*'.
> - **Errors on semantically ambiguous categories.** We believe this is largely due to inherent flaws in the benchmark annotations.  For example, several categories are conceptually close and overlap semantically (e.g., water/lake/river, building/skyscraper). Although VIP improves inter-class discrimination, such ambiguity can still cause segmentation errors.
>
> We will offer an in-depth failure analysis in our revision. We thank the reviewer again for raising this important question!
>
> -----
>
> > **`Q3: Extend to large-scale categories`**
>
> Thank you for this valuable question! The results in the main paper (**Table 1**) on ADE20K (**150 categories**) already suggest that VIP works well under large-scale categories. To further verify this, we conduct more experiments on **A-847** (847 categories) and **PC-459** (459 categories), reported in **Table-Class** ([`Anonymous link`](https://anonymous.4open.science/r/ICML_2320/Table-Class.md)). The results show that our method delivers notable gains on both benchmarks, supporting its strong generalizability.
>
> For computational cost, we would like to clarify that: 1) the alias distillation is fully parallelizable without complex iterative optimization; 2) the bottleneck lies in the image backbone rather than the text branch. As shown in **Table-Class**,  the overhead remains acceptable even in large-scale category settings.
>
> -----
>
> > **`Q4: Transferability of alias library`**
>
> Thank you for the insightful comment!  To address your concern,  we directly transfer the library distilled on VOC to overlapping classes in COCO-Object. Results are summarized in **Fig-Alias** ([`Anonymous link`](https://anonymous.4open.science/r/ICML_2320/Fig-Alias.pdf)), which suggest two findings: 1) for **semantically stable and unified** categories, the transfer works well. For instance, the alias library for the "person" distilled on VOC transfers seamlessly to the COCO-Object; 2) for domain-specific categories with **semantic shifts**, direct transfer is empirically less effective. For example, "table" in VOC specifically means "dining table", while in COCO it is far more general. In such cases, dataset-specific re-scoring is needed.

---

> > ### Author Rebuttal · Reviewer_KtmB · 2026-03-31
> >
> > The authors have adequately addressed my questions.

---

> > > ### Author Response · Authors · 2026-04-01
> > >
> > > Dear Reviewer `KtmB`,
> > >
> > > Thank you for your thoughtful feedback and support for our work! We greatly appreciate your constructive insights, particularly regarding failure case analysis, transferability of alias library, and extension to large-scale categories. Following your suggestions, we will incorporate additional discussions and analyses in the revised manuscript. We sincerely thank you once again for your insightful comments, which have helped us further strengthen our work.
> > >
> > > Best Regards,
> > >
> > > Authors of Paper 2320

---

### Official Review · Reviewer_drjx · 2026-02-25

**Soundness:** 2
**Presentation:** 3
**Significance:** 3
**Originality:** 2
**Overall Recommendation:** 4
**Confidence:** 5

**Summary:**

This paper focuses on the task of training-free open-vocabulary semantic segmentation, aiming to improve dense vision-language alignment. One key observation is that while frameworks like DINO mitigate spatial bias, they remain limited by cross-modal mismatch where spatially-aware visual features lack precise semantic anchors in the text modality. The VIP (Visual-guided Prompt Evolution) framework is proposed to leverage LLM-driven semantic expansion for category aliases, visual-guided alias distillation using visual priors for filtering, and saliency-aware soft aggregation for consensus mapping. Experiments are carried out on eight natural image benchmarks and four remote sensing datasets to assess mIoU, inference latency, and memory efficiency compared to CLIP-based and multi-model methods.

**Compliance With Llm Reviewing Policy:**

Affirmed.

**Final Justification:**

The authors have addressed most of my concerns, so I raise the rating to acceptance.

**Key Questions For Authors:**

Please see the weakness.

**Limitations:**

Yes

**Strengths And Weaknesses:**

[+] Important Task Identification: For many fine-grained deployments, dense vision-language alignment brings great value for the community.

[+] Clear Logic and Complete Manuscript: The paper is well-written with a logical flow. Images, tables, descriptions, and formulas are all appropriate.

[+] Extensive Evaluation: The paper evaluates performance across classic tasks, providing a comprehensive understanding of VIP’s advantages.

[-] Dataset-level Dependency in Distillation: The "Visual-guided Alias Distillation" scores aliases based on the average performance across the test dataset ("D represent the test set"). This raises concerns regarding the "zero-shot" nature for individual images. In a truly open-world, real-time scenario where only a single image is provided, the model would be unable to compute the VG Score or SC Score to filter aliases. The method currently behaves more like a "dataset-level prompt optimization" rather than an image-adaptive inference technique.

[-] Reliance on Frontier LLMs: The framework defaults to using GPT-5 (or latest frontier models) for semantic expansion. While effective, the paper lacks a sensitivity analysis regarding the LLM's capability. It is unclear if the "evolution" process remains robust when using smaller, open-source models (e.g., Llama-3-8B or Qwen-2.5-7B). Given that LLMs can hallucinate or provide semantically irrelevant synonyms, the dependence on high-end LLMs might limit the method's accessibility.

[-] Heuristic "Canonical Anchor" Assumption: The automated filtering mechanism assumes that the canonical class name provided by the dataset is a reliable "anchor" for scoring. However, in specialized domains like remote sensing or medical imaging, canonical names can be highly technical or obscure. If the anchor name itself yields a poor VG/SC score, the entire filtering logic (retaining aliases with higher VG and lower SC than the anchor) may collapse or lead to a degenerate alias library.

[-] Complexity of the Aggregation Module: While the "Saliency-aware Soft Aggregation" uses a free-energy function to probabilisticly merge activation maps, the manuscript provides limited theoretical or empirical justification for why this specific formulation is superior to simpler alternatives like learned weighted averaging. The jump from global average pooling to free-energy-based dense modulation involves several manually defined hyperparameters (e.g., α,β,τ) that may require per-dataset tuning to maintain the reported gains.

[-] Under-explored Text Template Transferability: While the authors claim transferability to text templates (§3.5), this section is treated as a "preliminary exploration." The paper would benefit from a more detailed meta-analysis comparing whether class-alias refinement or template-refinement contributes more to the final performance, as current benchmarks combine both without showing the individual Pareto frontiers for templates.

---

> ### Author Rebuttal · Authors · 2026-03-31
>
> > **`W1: Dataset-level dependency`**
>
> We sincerely thank the reviewer for the insightful feedback! To address your concern, we would like to clarify that:
>
> - Our approach is more akin to a **self-correction** process rather than a dataset-level optimization. As noted in **Appendix B.1 (Line 855-864)**, we can obtain the dense image-text segment maps for all aliases during the distillation, thus, the entire pipeline requires **only one single forward pass for each dataset**. Moreover, the whole process strictly follows the *training-free* paradigm, and is consistent with common practice for evaluating training-free methods on batches of target images.
>
> - To verify the generalization, we evaluate a single-image variant (**Table-Single** ([`Anonymous link`](https://anonymous.4open.science/r/ICML_2320/Table-Single.md))), where alias filtering is conducted independently for each image and then applied to the same image. It still **substantially outperforms the canonical baseline** (by 10.3 average mIoU), indicating that our approach remains highly effective even in a single-image setting.
>
> - We also conduct a split-transfer experiment  (**Table-Split** ([`Anonymous link`](https://anonymous.4open.science/r/ICML_2320/Table-Split.md))), where the alias library is distilled on the training split and directly deployed on the test split without re-scoring. This variant also works well, and on most datasets even surpasses building the library from the test split itself. This suggests that alias distillation can be performed on pre-collected samples and deployed directly on unseen data. More broadly, it reveals that our method is naturally **data-scalable**: as more unlabeled target data become available, alias ranking becomes more reliable and performance improves.
>
> ------
>
> > **`W2: Reliance on Frontier LLMs`**
>
> Due to space limits, the LLMs analysis was placed in **Appendix C.1 (Table 4)** of submitted manuscript. In addition, as noted in the **Appendix B.1 (Line 883-886)**, we adopt a simple strategy to filter hallucinated outputs from the LLM so that candidate aliases remain semantically close to the canonical class name.
>
> We also provide additional results on lighter open-source models (please see our response to **`#Vvdx-W3`**), which show that our method is insensitive to the specific LLM choice and works well across all models.
>
> ------
>
> > **`W3: Obscure canonical anchor`**
>
> In practice, our method is able to address this issue in most cases. To validate this, we provide detailed analyses of obscure anchors in the **Fig-Anchor** ([`Anonymous link`](https://anonymous.4open.science/r/ICML_2320/Fig-Anchor.pdf)).
>
> Specifically, when a canonical name is obscure (e.g., *"facade"* in VDD) or overly broad (e.g., *"vegetation"* in Cityscapes), the baseline yields a poor visual response and sets a low filtering bar. Thus, the expanded aliases that are visually concrete and discriminative can easily surpass this threshold. Our method successfully rectifies the flawed anchor with better text queries, driving remarkable gains (**+21.3%** for 'facade' and **+37.1%** for 'vegetation').
>
> We acknowledge that failures may occur in extreme cases, but such cases are rare in existing benchmarks and practical deployment. We will discuss this in the limitation of the manuscript.
>
> ------
>
> > **`W4: Justification of the aggregation module`**
>
> In a given image, not all aliases are expected to appear. Simple average pooling dilutes valid activations with noise from unactivated aliases, while max pooling is highly vulnerable to spurious outliers. Our free-energy formulation inherently resolves this by non-linearly rewarding collective consensus among activated aliases while suppressing isolated noise (as noted in **Section-3.4** of the main text). Furthermore, introducing "learned" weighted averaging violates our *training-free* setting, and naive heuristic weighting is empirically suboptimal (see empirical evidence in **Table-Aggre** ([`Anonymous link`](https://anonymous.4open.science/r/ICML_2320/Table-Aggre.md))).
>
> For hyperparameters, the results in the **Appendix C.1 (Figure 8a)** have demonstrated that our method remains stable across a wide range of values. Notably, as detailed in the **Appendix C.1 (Line 963)**, we use same hyperparameter settings for all benchmarks, with no dataset-specific tuning.
>
> ------
>
> > **`W5: Exploration of text template`**
>
> Thank you for the insightful comment! The core contribution of our work is to address the **cross-modal mismatch caused by canonical class names** in benchmarks. Our exploration of text templates is intended to show the broader applicability of text-side refinement, *rather than* to serve as a core component.  As shown in **Table 3** of the main paper, the major performance gains come from **class alias refinement and aggregation mechanisms**, while template refinement brings only a modest improvement. We will add a detailed discussion of text templates in the future work section.

---

> > ### Author Rebuttal · Reviewer_drjx · 2026-04-05
> >
> > The authors have addressed my concerns, so I raise the rating to acceptance.

---

> > > ### Author Response · Authors · 2026-04-06
> > >
> > > Dear Reviewer `drjx`,
> > >
> > >
> > >
> > > We sincerely thank the reviewer for the positive reassessment and for recognizing that our rebuttal has addressed all concerns! We greatly appreciate the constructive feedback provided during the review process, which has helped strengthen our work. We are encouraged by this outcome and remain committed to delivering a high-quality camera-ready version that incorporates your suggestions.
> > >
> > > Thank you again for your time and effort!
> > >
> > >
> > >
> > > Best Regards,
> > >
> > > Authors of Paper 2320

---

### Official Review · Reviewer_B93n · 2026-03-10

**Soundness:** 3
**Presentation:** 3
**Significance:** 2
**Originality:** 2
**Overall Recommendation:** 5
**Confidence:** 4

**Summary:**

This paper proposes VIP (Visual-guided Prompt Evolution), a method that improves the performance of dino.txt in training-free open-vocabulary semantic segmentation by carefully using the text prompts. Their method does not require any additional training on downstream tasks.

**Compliance With Llm Reviewing Policy:**

Affirmed.

**Final Justification:**

The rebuttal has adequately addressed my questions. Therefore, I decided to raise the score.

**Key Questions For Authors:**

The reviewer does not have major concerns, but I would be happy if the authors could address the following questions：
1.Could the authors provide results from some training-based OVSS methods using the weights they have shared? As an oracle-style comparison, this could help reviewers better assess the significance of the training-free OVSS approach.

2.Additionally, what would happen if the text decoder were replaced with a small language model? Would this improve the current experimental results, particularly when using textual descriptions?

**Limitations:**

Yes. This work does not appear to involve any obvious negative societal impacts.

**Strengths And Weaknesses:**

**Strengths**

1.The paper is clearly written and easy to follow. The motivation is clear, and the proposed framework is logically structured. The method description is concise yet sufficiently detailed, making the overall contribution easy to understand.

2.The paper identifies the gap between canonical class names and natural language descriptions in the pretraining stage as a key bottleneck in open-vocabulary semantic segmentation. The proposed visual-guided prompt evolution mechanism effectively leverages synonym information through LLM-based expansion and alias filtering, leading to consistent performance gains.

3.The authors conduct solid experimental evaluations, including detailed ablation studies to validate each component of the method. The appendix further evaluates VIP on multiple backbones, providing evidence of its generality. The method achieves strong performance across benchmarks while maintaining reasonable efficiency, compared to methods that utilize SAM.

**Weaknesses**

1.Although the method is training-free for downstream tasks, it still relies on substantial pretraining, including aligning DINOv3 with a text encoder. Therefore, the approach ultimately depends on strong pretrained components rather than being entirely free of supervision.

2.The proposed method inherits the limitations of the dual-tower architecture. Similar to CLIP-style models, both the text encoder and the overall cross-modal design remain relatively lightweight and lack deep semantic reasoning capabilities. As a result, the model may struggle to handle complex or compositional textual descriptions, limiting its effectiveness in more challenging open-vocabulary scenarios.

---

> ### Author Rebuttal · Authors · 2026-03-31
>
> > **`W1: Reliance on pretrained model`**
>
> Thank you for your valuable feedback! We would like to address this concern from two aspects:
>
> - In the training-free OVSS literature, the **standard setting** is to directly exploit the capabilities of a pretrained vision language model for text-specific segmentation without any retraining or finetuning. Our method strictly adheres to this paradigm by freezing all model weights and introducing no parameter updates during deployment.
>
> - Our method builds upon the dino.txt foundation, which is pretrained using only image **self-supervision** and **web-crawled image-text pairs**. In fact, the image-text supervision is **fundamentally necessary** to align visual and textual feature spaces. Given that web-scale image-text data is abundant and does not require dense human annotation, we believe that unlocking high-fidelity dense prediction capabilities of such models is **practically scalable**.
>
> ------
>
> > **`W2: Dual-tower architecture limitations`**
>
> To address your concern, we would like to clarify:
>
> 1. **Long text reasoning of dual-tower architectures:** Some works [1,2] show that CLIP-style VLMs can also be **substantially improved** for long-text understanding when the context length, pretraining data, and objective are redesigned appropriately. It suggests that the bottleneck is not purely architectural, and the dual-tower models can also achieve superior complex text reasoning without sacrificing the lightweight nature.
> 2. **Complex descriptions in dense prediction:** On the other hand,  for OVSS task, richer text is not always more helpful. As demonstrated by OpenSeg-R [3] and our ablation (**Appendix C.1, Table 5**), utilizing coarse category descriptors *degrades* segmentation performance. It indicates that class name queries act as **stronger, noise-free semantic anchors**, whereas course descriptions may introduce undesirable perturbations into dense image-text matching, unless subjected to fine-grained modulation.
>
> [1] Unlocking the long-text capability of clip. ECCV 2024.
>
> [2] Lotlip: Improving language-image pre-training for long text understanding. NeurIPS 2024.
>
> [3] Openseg-r: Improving open-vocabulary segmentation via step-by-step visual reasoning. arXiv 2025.
>
> ------
>
> > **`Q1: Results on training-based methods`**
>
> Thank you immensely for the instructive advice! To present a comprehensive comparison, we conduct additional experiments on training-based OVSS methods.
>
> The results are shown in **Table-Train** ([`Anonymous link`](https://anonymous.4open.science/r/ICML_2320/Table-Train.md)), we draw two key findings as follows. *First*, although our method requires no mask guidance or parameter updates, it remains highly competitive with training-based approaches, and even **outperforms** some of them. This highlights the strength of our training-free paradigm. *Second*, our proposed text evolution also works well when applied to training-based OVSS model. In particular, it improves FC-CLIP by 2.0 average mIoU, suggesting that our method extends beyond the training-free setting and generalizes to supervised baselines as well.
>
> We will add these results and discussion in our updated manuscript.
>
> ------
>
> > **`Q2: Replacing the text encoder`**
>
> Your keen insight is truly appreciated!  While replacing the text encoder with a small language model may enhance text understanding, it is **not** plug-and-play. Changing the text encoder **breaks** the pre-trained embedding space in dino.txt and requires large-scale re-training to align the new language model with the image encoder. This process demands substantial image–text data and training resources, making it infeasible in the short term.
>
> To address the reviewer’s concern, we choose to perform evaluation based on **LLM2CLIP**, which replaces the CLIP text encoder with **LLaMA 3.1 8B**. The results are shown in **Table-LLM2CLIP** ([`Anonymous link`](https://anonymous.4open.science/r/ICML_2320/Table-LLM2CLIP.md)). We find that although LLM2CLIP improves the model’s ability to process long-form text, using category textual descriptors still leads to a performance **drop** in OVSS. Even in the original LLM2CLIP paper, long-form descriptions are only used for image-text retrieval, while class name queries is the default choice for classification and OVSS.
>
> These findings further support our point in **W2 response**: uncurated class descriptors tend to introduce unexpected noise, while class name queries provide a more robust semantic anchor for OVSS. In addition, we show that our method also brings consistent gains when applied to LLM2CLIP, validating the broader applicability of our core idea. We will incorporate a detailed discussion into the manuscript.
>
> [1] LLM2CLIP: Powerful Language Model Unlocks Richer Cross-Modality Representation. AAAI 2026.

---

> > ### Author Rebuttal · Reviewer_B93n · 2026-04-03
> >
> > The authors have addressed my questions.

---

> > > ### Author Response · Authors · 2026-04-03
> > >
> > > Dear Reviewer `B93n`,
> > >
> > >
> > > Thank you for your thoughtful feedback and for taking the time to review our responses. We truly appreciate your constructive insights and your willingness to engage with our clarifications. Your support of our work, especially **the recognition of our motivation to bridge the vocabulary gap and the thoroughness of our empirical validations**, means a lot to us. We promise to open-source our code on all benchmarks in the future to inspire subsequent research.
> > >
> > >
> > > Best Regards,
> > >
> > > Authors of Paper 2320

---

### Official Review · Reviewer_Vvdx · 2026-03-15

**Soundness:** 3
**Presentation:** 3
**Significance:** 3
**Originality:** 2
**Overall Recommendation:** 4
**Confidence:** 3

**Summary:**

This paper presents VIP, a training-free open-vocabulary semantic segmentation method built on top of dino.txt. The paper argues that dino.txt provides stronger spatially-aware dense features than CLIP-based approaches. However, its segmentation performance is still limited by the semantic mismatch between canonical class names and the language distribution seen during pretraining. To address this issue, the authors propose VIP, which uses an LLM to generate class aliases, filters them using visual-grounding and semantic-certainty scores derived from visual priors, and softly aggregates the retained activations. Experimental results are reported on benchmarks spanning natural images, urban street scenes, and remote sensing imagery.

**Compliance With Llm Reviewing Policy:**

Affirmed.

**Final Justification:**

My concern is mostly addressed so I retain my positive rating.

**Key Questions For Authors:**

1. Could the authors clarify which backbone is actually used?
2. Could the authors clarify the inference pipeline for unseen/in-the-wild categories, and provide an estimate of the runtime cost for alias generation and filtering?

**Limitations:**

yes

**Strengths And Weaknesses:**

Strengths

1. The paper is clearly written, with a well-structured method section and a clear motivation for each module.

2. The proposed filtering metrics are interesting and well motivated. In particular, the visual grounding score and semantic certainty score form a reasonable pair for selecting useful aliases.

3. The empirical evaluation is comprehensive, covering multiple standard OVSS benchmarks as well as remote-sensing datasets.

Weaknesses

1. There appears to be an inconsistency regarding the backbone underlying dino.txt. The paper repeatedly describes dino.txt as using DINOv3, while the cited dino.txt reference is “Dinov2 meets text,” making it unclear whether VIP is built on the original dino.txt or a DINOv3 retrained variant. This should be clarified.

2. The practicality of the alias-expansion pipeline for truly open-world categories is unclear. The method uses GPT-5 to generate candidate aliases, and the filtering scores appear to be computed over a dataset (Sec 3.3). It would therefore be helpful to clarify the deployment procedure for unseen/in the wild categories, report the additional cost of alias generation/filtering.

3. It would be helpful to also include results with alias generated using an open-source LLM instead of GPT-5.

Question / suggestion

Since DINOv2/DinoV3 backbones are trained with register tokens, and prior work suggests these tokens capture high-level semantic information, it would be interesting to discuss whether register tokens could be used as an alternative visual prior for query refinement as affinity matrix has been explored by many prior works.

---

> ### Author Rebuttal · Authors · 2026-03-31
>
> > **`W1&Q1: Clarification of backbone`**
>
> We thank the reviewer for this comment! We have already clarified in **Appendix B.1 (Lines 844-846)** that although the original dino.txt paper utilized DINOv2, their official open-source code was *updated* to support DINOv3 backbone. All results reported in our paper are built upon the latest version (including baseline) for fair comparison. We will update the manuscript to make this clarification explicit in the main text.
>
> ------
>
> > **`W2&Q2: Clarification of inference pipeline`**
>
> Thanks for your valuable feedback! To address your concern, we would like to clarify that:
>
> - Given a test benchmark, our approach generates aliases for all canonical class names and utilizes the benchmark images for alias library distillation and aggregation in a fully training-free manner. In practice, we can obtain the dense image-text segment maps for all aliases during the distillation process, thus, the entire pipeline requires just **one single forward pass** for each dataset. As stated in **Appendix B.1 (Lines 855-864)**, the inference cost in Table 1 of the main paper reports the average per-image overhead for this complete pipeline, which accurately evaluates real-world computational demands.
>
> - We additionally evaluate a single-image variant, where alias filtering is performed independently for each image and then applied to the same image for testing, as presented in **Table-Single** ([`Anonymous link`](https://anonymous.4open.science/r/ICML_2320/Table-Single.md)). This variant remains **substantially better than the canonical baseline** (by 10.3 average mIoU), demonstrating that our approach remains highly effective even in a single-image setting.
>
> - We also perform an additional split-transfer experiment in **Table-Split** ([`Anonymous link`](https://anonymous.4open.science/r/ICML_2320/Table-Split.md)): we distill the alias library on the training split and then directly deploy it on the test split without re-scoring. This variant also works well, and on most datasets it even performs **better than** building the library from the test split itself. We believe this is a strong indication that suggests a practically useful usage pattern of VIP: one can score aliases once on a modest unlabeled data pool and then deploy the resulting library directly. More broadly, the results also reveal our framework is naturally **data-scalable** and works in a **self-correction** paradigm: as the amount of unlabeled target data increases, alias ranking becomes more reliable.
>
> ------
>
> > **`W3: More results on open-source LLMs`**
>
> We apologize for not making this clearer in the main text.  We have already provided ablation studies on LLMs in **Appendix C.1 (Table 4)**, which show that replacing GPT-5 with Gemini-2.5 or DeepSeek-R1 works equally well. In addition, we provide more results in **Table-LLM** ([`Anonymous link`](https://anonymous.4open.science/r/ICML_2320/Table-LLM.md)), including comparisons with lighter open-source models such as Qwen-2.5-7B and Llama-3-8B. The results clearly show that the performance of our VIP framework is highly **insensitive** to the choice of LLM.
>
> Intuitively, generating diverse aliases is a *basic* capability of modern LLMs, and our method does **not require** more complex query reasoning beyond this. Therefore, the choice of LLM has only a limited impact. Moreover, our ablations show that semantic expansion via LLM alone fails to bring gains, and instead, the improvements are largely attributed to our **alias refinement and aggregation mechanisms**.
>
> We will move this analysis to the main text to make this point clearer.
>
> ------
>
> > **`Suggestion 1: Discussion on register tokens`**
>
> The reviewer's idea coincides exactly with our preliminary explorations! We agree that high-level semantics from register tokens hold great potential for training-free OVSS. In our early experiments, we attempted to utilize the global semantics of the [CLS] token to compute global activation scores and classification entropy for alias filtering, but **this did not work**. We have provided these findings, along with additional experiments using *register tokens* as global semantic guidance, in the **Table-Global** ([`Anonymous link`](https://anonymous.4open.science/r/ICML_2320/Table-Global.md)).
>
> The results show that introducing global cues from CLS/register tokens into our method leads to a *drop* in performance. We attribute this mainly to the fact that dense prediction models are highly sensitive to text queries, and incorporating global cues may introduce substantial unexpected noise, thereby degrading performance.
>
> On the other hand, our method focuses on exploiting the patch-level pairwise structural priors in DINOv3. From this perspective, the global semantic serve as *complementary* cues for query refinement. A deeper investigation of this direction would be a worthwhile avenue for future work. We will include a more detailed discussion in the revised manuscript.

---

> > ### Author Rebuttal · Reviewer_Vvdx · 2026-04-05
> >
> > The authors have adequately addressed my concerns and also answer my question with detail.

---

> > > ### Author Response · Authors · 2026-04-06
> > >
> > > Dear Reviewer `Vvdx`,
> > >
> > >
> > >
> > > Thank you for your positive feedback and for acknowledging that our rebuttal successfully resolved all your concerns! We truly appreciate your meticulous review and high-caliber suggestions, including **insights of high-level semantic information for query refinement**. These have significantly enriched the depth and quality of our manuscript. It has been an honor for us to incorporate your recommendations and suggestions into our revised manuscript.
> > >
> > > Thank you once again for your time, expertise, and constructive review!
> > >
> > >
> > >
> > > Best Regards,
> > >
> > > Authors of Paper 2320

---

### Decision · Program_Chairs · 2026-04-30

**Decision:**

Accept (regular)

**Comment:**

All reviewers are positive for this paper. The main strengths include: 1) The proposed method is interesting and well motivated; 2) strong empirical results and generalization; 3) the paper is well written. Although there were some concerns raised by the reviewers, they were adequately addressed by the rebuttal. The The AC agrees with the reviewers, and decides to accept this paper.